



# CCN characteristics during the Indian Summer Monsoon over a rain-shadow region

Venugopalan Nair Jayachandran[1], Mercy Varghese[1], Palani Murugavel[1], Kiran S. Todekar[1], Shivdas P. Bankar[1], Neelam Malap[1], Gurunule Dinesh[1], Pramod D. Safai[1], Jaya Rao[1], Mahen Konwar[1], Shivsai Dixit[1], Thara V. Prabha[1],

[1]Indian Institute of Tropical Meteorology, Pune, India.

*Correspondence to:* V. Jayachandran (jayanspl@gmail.com)

**Abstract.** Continuous aerosol and Cloud Condensation Nuclei (CCN) measurements carried out at the ground observational facility situated in the rain-shadow region of the Indian sub-continent are illustrated. These observations were part of the Cloud-Aerosol Interaction Precipitation Enhancement EXperiment (CAIPEEX) during the Indian Summer Monsoon season (June to September) of 2018. Observations are classified as dry-continental (monsoon break) and wet-marine (monsoon active) according to air mass history. CCN concentrations measured for a range of supersaturations (0.2-1.2 %) are parameterized using Twomey's empirical relationship. CCN concentrations even at low (0.2 %) supersaturation (SS) were high (>1,000 cm$^{-3}$) during continental conditions associated with high black carbon (BC~2,000 ng m$^{-3}$) and columnar aerosol loading. During the marine air mass conditions, CCN concentrations diminished to ~350 cm$^{-3}$ at 0.3 % SS and low aerosol loading persisted (BC~900 ng m$^{-3}$). High CCN activation fraction (AF) of ~0.55 (at 0.3 % SS) were observed before the monsoon rainfall, which reduced to ~0.15 during the monsoon and enhanced to ~0.32 after that. Mostly mono-modal aerosol number-size distribution (NSD) with a mean geometric mean diameter (GMD) of ~85 nm, with least (~9 %) contribution from nucleation mode (<30 nm) particles persisted before monsoon, while multi-mode NSD with ~19 % of nucleation mode particles were found during the monsoon. Critical activation diameters ($d_{cri}$) for 0.3 % SS were found to be about 72, 169, and 121 nm prior, during and, after the monsoon, respectively. The estimated $d_{cri}$ were inversely correlated to the AF, linearly up to 100 nm and non-linearly beyond that. The better association of CCN with aerosol absorption, and the concurrent accumulation mode particles during continental conditions, point to the possibility of aged (oxygenated) carbonaceous aerosols enhancing the CCN activity. The enhancement in CCN concentration and k-values during daytime along with the increase in absorption Angstrom exponent, indicate the freshly emitted local anthropogenic aerosols dominated by organics reducing the CCN AF during the monsoon. Best closure obtained using measured critical diameter, and ammonium sulfate composition during continental conditions emphasize the role of aged aerosols contributing to the accumulation mode, enhancing the CCN efficiency. But the over-estimation of CCN and less hygroscopicity of accumulation mode aerosols during the monsoon point to the significant role of size-dependent aerosol composition in CCN activity during the period.





# 1 Introduction

Atmospheric aerosol particles (AP) emitted from both natural and anthropogenic sources, affect the radiation budget as well as the hydrological cycle of Earth, mainly through its direct and indirect effects. Cloud Condensation Nuclei (CCN) the sub-set of AP or Condensation Nuclei (CN), that can be activated at a specific water vapor supersaturation (SS), and indirectly affect the climate by altering cloud micro-physical properties. For a fixed liquid water content, an increase in CCN concentration increases the cloud droplet number concentration (Twomey and Warner, 1967), while reducing the cloud droplet size (Twomey, 1974), and thereby altering the cloud albedo (Twomey, 1977) and lifetime (Albrecht, 1989). All these effects eventually modify the precipitation pattern (Lohmann and Feichter, 2005; Rosenfeld et al., 2008). Some of these aerosol indirect effects are moderately understood, while others are not, which contribute to significant uncertainty among all the climate forcing mechanisms (IPCC, 2013). Characterization of CCN is the most fundamental challenge in assessing the aerosol-cloud interactions (ACI) and reducing the associated uncertainties in indirect radiative forcing assessment. In this regard, the primary aspect is to characterize the hygroscopic growth of AP with respect to relative humidity, which is generally addressed by the Köhler theory (Köhler, 1936) and direct bearing from the physical and chemical characteristics of AP. However, Köhler theory is modified to accommodate the real atmospheric conditions and applied for both laboratory and field measurements, as well as in the climate models (Shulman et al., 1996; Laaksonen et al., 1998; Raymond and Pandis, 2003; McFiggans et al., 2006; Rose et al., 2008; Mikhailov et al., 2009).

Large spatial and temporal heterogeneities are found in AP as well as CCN properties and thus, the regional characterization of CCN in different meteorological settings are imminent. Temporal and spatial heterogeneities of CCN and different mechanisms affecting CCN are investigated in several studies (Hoppel et al., 1973, Hudson and Xie, 1999, Paramonov et al., 2015, Schmale et al., 2018) over both continental and marine environments. Over the land mass, significant variability in CCN activation properties are reported due to urban and industrial influences (Sotiropoulou et al., 2007; Asa-Awuku et al., 2011). For a given particle, the size and composition determine its CCN activity at a specific SS, while the CCN spectrum (CCN at different SS) depends on the median diameter and standard deviation, number concentration, and the mixing state of the aerosol system (Quinn et al., 2008). In this regard, closure studies are necessary to understand the role of each parameter in the activation of AP as CCN, which may improve the accuracy of climate models to address the ACI (Fountakis and Nenes, 2005). However, the role of organics, mostly from carbonaceous combustion sources, in determining the CCN activity is still uncertain. Ervens et al. (2005) have reported a broad range (-86 % to 110 %) of changes in cloud droplet number concentration due to the organics. The increasing trend in aerosol loading (Babu et al., 2013) and the significant contribution of carbonaceous aerosols from both fossil fuel and biomass burning over the Indian subcontinent (Nair et al., 2007) highlight the necessity of the characterization of CCN over distinct environments in India and the role of carbonaceous AP.

Even though aerosol properties such as aerosol optical depth (Babu et al., 2013), black carbon (BC) mass concentration (Manoj et al., 2019) have been studied across the Indian sub-continent through a network of observatories (Moorthy et al., 2013) for decades, only a few CCN studies are availbale since last few years over specific regions. Year-round



CCN measurements are reported from Western Ghats (Leena et al., 2016), Indo-Gangetic Plain (IGP, Patidar et al., 2012),
Central Himalayas (Gogoi et al., 2015), and Eastern Himalayas (Roy et al., 2017). CCN characteristics for a specific season,
including closure analysis were reported by Jayachandran et al., (2017; 2018) at peninsular India, and by Bhattu and Tripathi,
(2015) at IGP. Apart from these, Indian Ocean EXperiment (INDOEX, Ramanathan et al., 2001), Cloud-Aerosol Interaction
Precipitation Enhancement EXperiment (CAIPEEX, Kulkarni et al., 2012), and South-West Asian Aerosol-Monsoon
Interaction - Regional Aerosol Warming EXperiment (SWAAMI-RAWEX, Jayachandran et al., 2019), Integrated Campaign
for Aerosols, gases and Radiation Budget (ICARB-2018, Nair et al., 2019) are other major multi-platform campaigns carried
out over the sub-continent and nearby marine environment to study the regional ACI. CAIPEEX conducted both aircraft and
ground-based observations of aerosols, clouds, and planetary boundary layer (PBL) since 2009, in a phased manner. Details
of the CAIPEEX are available in Prabha et al., (2011) and Kulkarni et al., (2012). Various studies addressing spatio temporal
distribution of AP (Padmakumari et al., 2013; Varghese et al., 2019), cloud microphysics (Prabha et al., 2011; 2012;
Padmakumari et al., 2018), rainfall (Maheshkumar et al., 2014) properties, relationship between cloud microphysics and
thermodynamics (Bera et al., 2019), and ACI (Pandithurai et al., 2012; Prabha et al., 2012; Konwar et al., 2012; Gayatri et al.,
2017; Patade et al., 2019) from the unique data obtained from the CAIPEEX. Varghese et al., (2016) investigated the linkages
of surface and cloud base CCN spectral characteristics over the rain shadow region.

The assessment of the effects of AP on clouds and precipitation due to the changes in the atmospheric composition
by anthropogenic activities is very significant over India as the agriculture and economy of the region mostly depend on the
Indian Summer Monsoon (ISM) rainfall. The west coast of India, which is the gateway of the ISM, receives almost 2.5 times
the long-term monsoon mean rainfall observed all over India (Parthasarathy et al., 1995). The mountain ranges along the
western coast of India known as the Western Ghats (WG) mountains, play a pivotal role in ISM rainfall  due to orography
(Grossman and Duran, 1984; Sijikumar et al., 2013). WG mountain range is oriented in the north-south direction and extends
about 1,600 km and has an average altitude of about 1.2 km. A few studies have been carried out till date to understand the
aerosol loading (Udayasoorian et al., 2014), CCN characteristics (Leena et al., 2016; Jayachandran et al., 2018) and its
influence on the aerosol indirect effects (Anil Kumar et al., 2016) from different locations in the WG. It is observed that the
orography and the associated atmospheric dynamics cause heavy rainfall ($\sim$500 cm yr$^{-1}$) over the windward side of the WG,
while the rain shadow region (leeward side) is prone to drought conditions. CAIPEEX observations were conducted over the
rain shadow region to understand the cloud and precipitation microphysics and AP properties to derive guidelines for the
precipitation enhancement over the region. CAIPEEX Phase IV was designed to address the major objectives for the science
of weather modification. The background observations of CCN were trivial for the design and validation of the experiment
and the data presented in this study is aimed at understanding the aerosols and its cloud activation properties near the surface.
The present study addresses CCN and its characteristics under different air mass and meteorological conditions throughout the
ISM season over a rain shadow region, and was not available in earlier studies. The study focuses on the variations in CCN
characteristics within the ISM, and the possible factors are investigated using the concurrent and collocated aerosol



measurements. CCN closure analysis is carried out to assess the role of size and composition at different atmospheric conditions.

## 2 Experiment details

### 2.1 Location, measurements, and database


As part of the ground segment of CAIPPEX IV campaign, aerosol and PBL measurements have been going on since May 2017 from Sinhgad College of Engineering at Solapur (17.70° N, 75.85° E, ~490 m a.m.s.l), which is at the core of the rain shadow region. The location is marked as a circle in Figure 1 and is a semi-arid region. The college is at the Solapur city outskirts (~12 km away) and the aerosol sampling lab is on the third floor, away from all local activities in the rural setting.

Even though the sampling site is well isolated from the urban contamination, Solapur consists of numerous sugar and textile industries that emit smoke, apart from the seasonal emissions from agricultural activities.

Details of instrumentation and data used for the present study is illustrated in Table 1. Aerosol sampling was carried out through separate PM 2.5 inlets from about 2 m above the rooftop connected with conductive tubing. CCN concentrations were measured at every second using a continuous flow streamwise temperature gradient CCN counter (CCN-100, DMT Inc; 110 Roberts and Nenes, 2005). Initially (June 2018) CCN were measured at five SS (0.2, 0.4, 0.6, 0.8 and 1.0 %) and in July CCN counter was calibrated again and the SS was set at 0.3, 0.5, 0.8 and 1.2 %. Calibrations were carried out both before and after the experiments following Rose et al., (2008). CCN counter uses the fundamental principle of the difference in the diffusion rate of heat and water vapor. A fixed temperature gradient is maintained along the walls of the wetted cylindrical column inside the instrument in which the desired SS is generated depending on the temperature gradient and the flow rate. The aerosols are 115 fed at a constant sheath to sample flow of (10:1) along the center-line of the column and the total flow rate was maintained at 500 Vccm. The details of the working principle of the instrument are available in Roberts and Nenes, (2005) and Lance et al., (2006). During June, each SS was maintained for five minutes, except for 0.2 % which was for 10 minutes. About two minutes of data during the SS transition were discarded to avoid the uncertainty in establishing the required SS during the transition. At 0.2 % SS (lowest set-SS), about four minutes of initial data were discarded. Except June, all the SS were set for seven 120 minutes each, except for 0.3 % SS, which was maintained for nine minutes. Here also, the initial 3-4 minutes data were discarded to ensure the set SS conditions. Thus, one cycle of the complete set of SS took 30 minutes, and the cycle was repeated. AP were continuously exposed to the SS inside the column, and those having their critical SS less than that of the set-SS inside the column, activated as liquid droplets and counted by the optical particle counter operated by a laser diode at 660 nm at the exit of the column. Since the CCN concentration was always less than 6,000 cm$^{-3}$, correction for water vapor 125 depletion inside the column as suggested by Lathem and Nenes, (2011) was not applied.

Size segregated aerosol number concentration (NSD) from about 15 nm to about 685 nm, distributed among 107 size bins, was measured every three minutes using a scanning mobility particle sizer (SMPS, TSI model 3082). The set up consists of an electrostatic classifier, including a long Differential Mobility Analyser (LDMA, TSI model 3081), and a butanol based-



Condensation Particle Counter (CPC, TSI model 3772). Before entering the LDMA the AP are charged to a known charge
130    distribution by a bipolar charger in the electrostatic classifier, which were size segregated according to their electrical mobility
(Wiedensohler, 1988; Wang and Flagan, 1990) in the DMA. The AP classified according to their sizes were counted by the
CPC. The sheath and sample flow were maintained at 0.3 and 3 L min⁻¹, respectively. Multiple charge correction and diffusion
charge correction were applied to the aerosol NSD data during the data inversion. AP were passed through a diffusion dryer
before the classifier to prevent the high humidity conditions.

135    Radiation absorption properties of AP at different wavelengths were measured using a dual spot Aethalometer (AE
33, make: Magee Scientific) at every minute. Aethalometer operated at a flow rate of 2 LPM measured the attenuation of light
due to the AP deposited on a filter tape (Hansen et al., 1984) at seven different wavelengths - 370, 470, 520, 590, 660, 880 and
950 nm. From this, the absorption coefficient ($\sigma_{abs}$) is estimated from the rate of attenuation, filter spot area, and the flow rate
(Weingartener et al., 2003). The new-generation AE33 compensates the loading effect and multiple scattering effects (Arnott
et al., 2005) associated with the filter-based optical attenuation techniques (Drinovec et al., 2015).
The wavelength dependence of absorption coefficient of aerosols is parameterised using the equation

$$\sigma_{abs}(\lambda) \;=\; \beta \,\times\, \lambda^{-\alpha_{abs}} \qquad\qquad (1)$$

where, $\beta$ is a constant and $\alpha_{abs}$ is the absorption Angstrom exponent. The nature of the carbonaceous sources can be inferred
from the value of $\alpha_{abs}$. Humic-like substance (HULIS) and brown carbon produced from biomass burning have higher
absorption at lower wave length (ultra violet and blue) regions (Gelencser et al., 2003). Hence, $\alpha_{abs}$ will be higher (~2) for
biomass dominant sources, while fossil fuel dominant sources will have $\alpha_{abs}$ close to unity (Kirchstetter et al., 2004)

Ambient weather parameters such as temperature, pressure, wind speed, wind direction, relative humidity and rainfall
were also used in the present study from the Automatic Weather Station (AWS) measurements located at the site. All the
instruments operated during CAIPEEX were calibrated periodically, especially before and after the experiments. All the
measurements having different sampling frequencies were averaged to hourly intervals for analysis and interpretations. Air
mass pathways were investigated using Hybrid Single Particle Lagrangian Integrated Trajectory Model (HYSPLIT) (Draxler
and Rolph, 2014) available from NOAA ARL READY Website.

**2.2 Meteorology**

The air mass back trajectories for five days reaching 50 m above the site were examined using back-trajectory analysis
and found that two distinct air masses reached the site during the observation period. These are classified as (a) continental
(dry) and (b) marine (wet), and shown in Figure 1. Continental classification is carried out for those air masses which were
over the landmass and within 1 km a.g.l for minimum 3 days before reaching the site, and hence have a significant continental
influence. All other trajectories, which were from the nearby marine atmosphere are classified as the marine and the
observation period includes the monsoon rainfall period over the site. Continental air mass consistently prevailed over the site
during the first week of June and from September 15 of 2018. The ISM onset over Solapur was on 8 June 2018. Monsoonal
circulation consistently prevailed over the site during July and August months of 2018. Thus, the observations and findings





throughout this manuscript are examined on the basis of this classification. It can be seen from Figure 1(a) that most of the cases, air masses were confined within lower 2 km, while typical monsoon circulation was reaching the site during the July and August months (Fig. 1b).

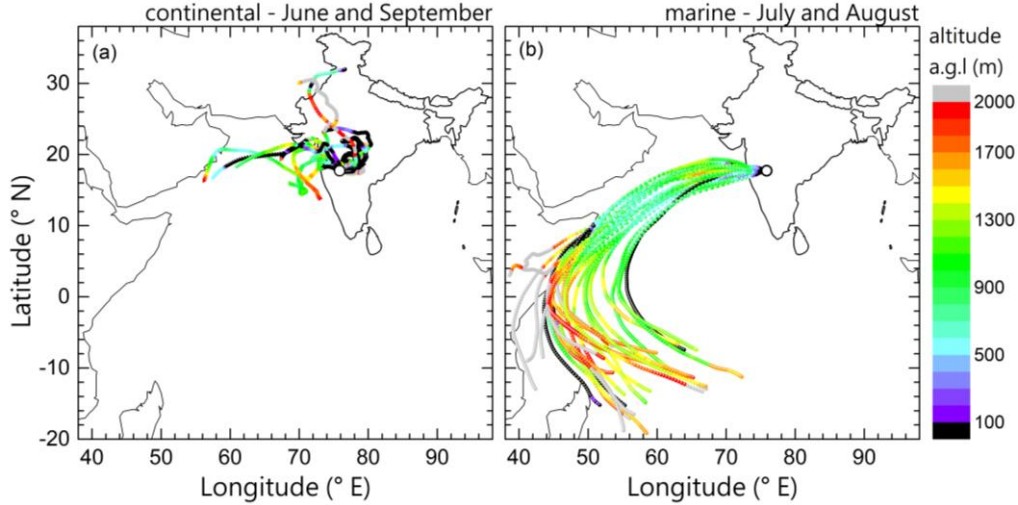

**Figure 1.** Air mass back trajectories with 5 days duration reaching 50 m above the site segregated to continental and marine. The color of the trajectories indicates the altitude of the air mass above the surface.

The meteorological parameters observed at the site from the AWS measurements during these periods are shown in Figure 2. Diurnal variation of temperature and relative humidity (RH) are shown in Figure 2(a) and (b), respectively. The temperature and RH values are distinctively different prior to the monsoon (June), compared to other periods. Monsoon (July and August) months experienced low temperatures and high RH throughout the day, while September had higher temperature and lower RH during noon and afternoon hours. The monthly mean temperature during the study periods of June, July, August, and September were 29.5±3.6°C, 25.9±2.6°C, 25.4±2.8°C, and 27.1±3.5°C; respectively. A dry spell existed during the campaign days of June when the maximum hourly mean temperature recorded was ~38 °C. The maximum temperature at all the months was observed at the 15 and 16 hours (IST), and the lowest temperatures were observed before sunrise of the day. Intermittent rainfall happened during July and August months and a few heavy rainfall events occurred during these months. The aerosol/CCN measurements during heavy rainfall are not included in the analysis for interpretations (missing days in Table 1). From the wind rose diagram (Fig. 2c), it can be noted that the strong winds were blowing mostly from the west and south-west part of the site during June and in few cases, winds were blowing from the north-east direction. Winds were reaching the site from west and south-west during July and August months (Fig. 2 d and e), while September had winds mostly from North-East and South-West with significantly weak wind conditions.

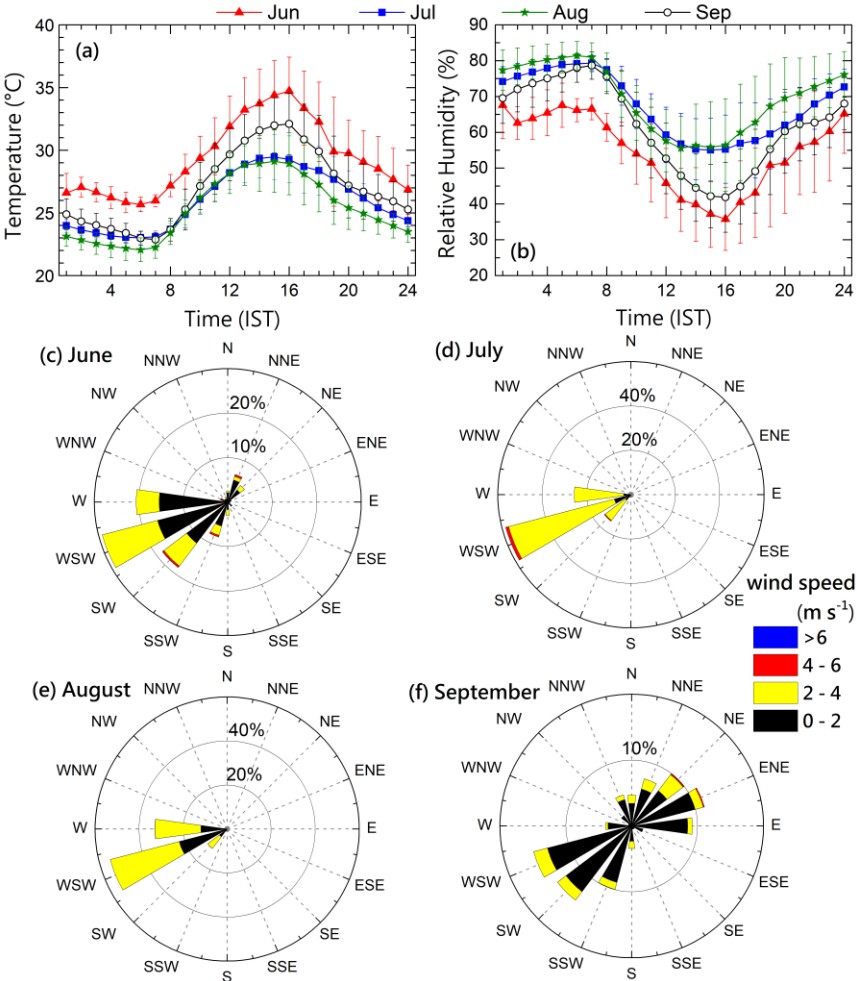

**Figure 2.** Diurnal variation of (a) temperature, (b) relative humidity along with the standard deviation of the mean values. The wind rose diagrams from co-located AWS measurements for the months of (c) June, (d) July, (e) August, and (f) September.

## 2       Results and Discussions

CCN characteristics at the site is investigated with aerosol size distribution and BC measurements.

### 3.1 Overview of aerosol loading

The frequency distribution of BC mass loading and the mean values (and its standard deviation) during the observation days are shown in Figure 3(a) with an aim to understand aerosol loading and the influence from anthropogenic activities. The distinct atmospheric conditions and the air mass history are evident in the BC mass loading at the site. Before the onset of monsoon, under the influence of continental air mass, the mean BC values were above 2,000 ng m$^{-3}$, which reduced to very





low values (~746 ng m⁻³) during the wet conditions. BC concentration was even higher than 4,000 ng m⁻³ during the continental air mass, while in many cases values were almost 100 ng m⁻³ under monsoon conditions.

Apart from the near-surface measurements, the columnar aerosol optical depth (AOD) is examined using the Moderate Resolution Imaging Spectroradiometer (MODIS) - Aqua at 550 nm. The AOD observed from MODIS before the monsoon onset and during the continental air mass conditions along with the site (white star) are shown in panels (b) and (c) of Figure 3. It can be seen that heavy aerosol loading (AOD>0.5) persisted around the rain shadow regions and the Mumbai coast (northwest of the site) in addition to the high loading over the IGP. After the monsoon rainfall, the aerosol loading has

reduced all over India as seen in Figure 3(c). Still, high aerosol loading (AOD>0.4) was observed around the observation site, IGP and the northern part of the east coast.

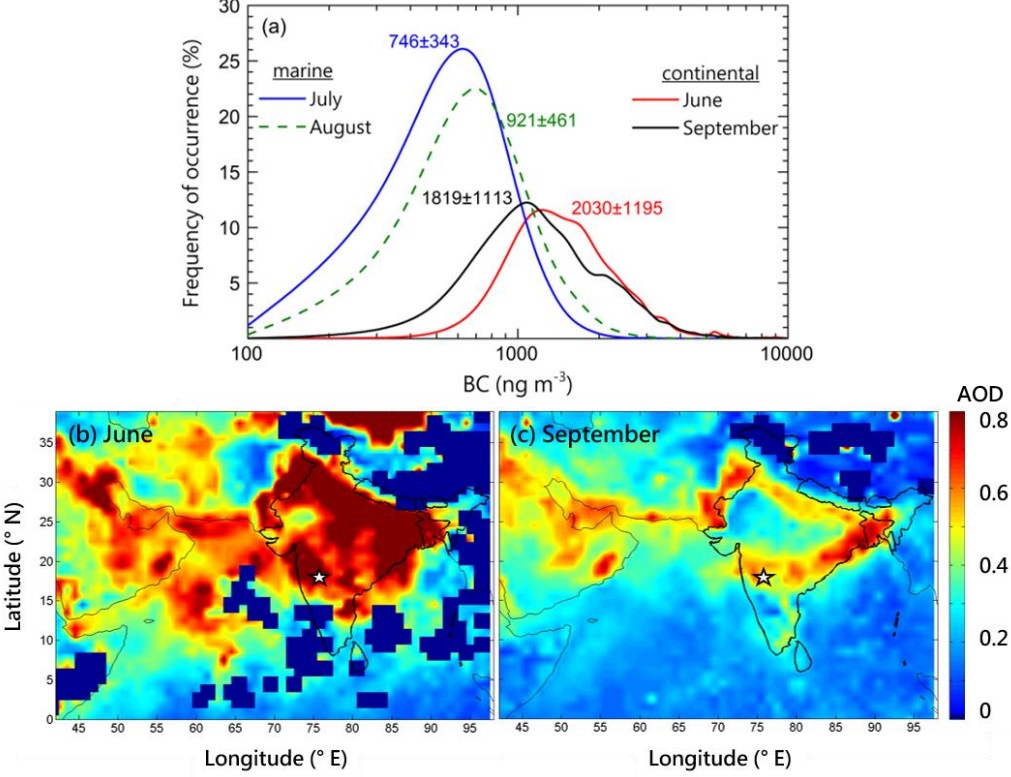

**Figure 3.** (a) Frequency distribution of BC mass concentration and its mean values for marine and continental conditions. Aerosol optical depth (AOD) at 550 nm observed from Moderate Resolution Imaging Spectroradiometer (MODIS) during (b) prior to (June) and (c) after

(September) monsoon, and the white star indicate Solapur.

BC can be considerred as a proxy for the anthropogenic activities (Myhre et al., 2013; Lelieveld et al., 2019), and the BC loading over Solapur indicates that the anthropogenic influence is predominant during the continental air mass conditions. This observation is in line with the columnar aerosol observation from MODIS. Apart from fossil fuel combustion, biomass

burning may also contributes to the carbonaceous aerosols that prevailed over the site. The fire counts observed from the



MODIS (collection 6 product obtained from https://earthdata.nasa.gov/firms) may support this inference, which is given in the Appendix (Figure A1). The high aerosol loading locations in Figure 3 are associated with the numerous fire events which can be seen in Figure A1. From another site in the rain shadow region closer to the central part of India - Nagpur, Kompalli et al., (2014) have reported BC mass of ~2,000 ng m$^{-3}$ before the monsoon, which is similar to the present study. From the long-term

observations of BC from the north-west part (Pune) of the current study, Safai et al., (2013) have reported a mean BC mass of ~1,200 ng m$^{-3}$ during the monsoon period. Both the high surface BC and total column aerosol loading observed before the monsoon, indicate the significant anthropogenic influence on the total aerosol loading. The low BC values (<1 μg m$^{-3}$) during the wet-monsoon months at Solapur represent a cleaner atmosphere, while ~1.5 μg m$^{-3}$ BC was reported from a coastal location in peninsular India (Babu and Moorthy, 2002) during monsoon. About 50 % reduction (10 to 6 μg m$^{-3}$) in BC mass associated

with the dominance of fossil fuel source replacing the biomass, during monsoon compared to the pre-monsoon values, was reported by Vaishya et al., (2017) from a heavily polluted IGP site. The low BC loading during the monsoon months over Solapur is due to the wet scavenging of aerosols and the distinct air mass reaching the site as well as due to less local burning during the active monsoon conditions. The high AOD and BC observations identify Solapur as a polluted-continental environment, which is cleaner during active monsoon compared to the other periods .

**3.2 CCN number concentrations and its variations**

The mean CCN concentration at different SS, known as the CCN SS spectra, segregated according to the air mass conditions are shown in Figure 4. It can be seen that the CCN concentrations at all SS are higher during June and September, compared to monsoon months (July and August). The highest CCN concentration is observed during June which is as per the surface BC loading and the total columnar aerosol loading. CCN spectra are similar for the monsoon months, except the slight

difference at the lowest SS. CCN concentration before the monsoon ranged from ~1,600 to 3,600 cm$^{-3}$ for 0.2 to 1.0 % SS. Meanwhile, the CCN concentration was only ~900 cm$^{-3}$ during July and August, even at 1.2 % SS. Thus, a clear distinction is seen in the CCN concentration between the wet and dry conditions within the same ISM period.

The CCN concentration varies with SS, and its parameterization is very important for its applicability in climate models (Khvorostyanov and Curry, 2006). The measured CCN spectra are parameterized by the Twomey's empirical fit

relationship (Twomey, 1959; 1977), which is widely used due to its simplicity (Cohard et al., 1998) and given as,

$$CCN(ss) = C \times SS^k \qquad (2)$$

where C and k are the empirical fit parameters characterizing the spectra.

More than 90 % of cases of the current observations show a high correlation coefficient (R>0.95) with the Twomey's empirical fit, except during September, during which about 65 % of the cases only had high (> 0.95) correlation coefficient

with the Twomey's fit. The spectra having a correlation coefficient of more than 0.95 with the empirical fit are only considered in the present study.

Hygroscopic or bigger particles have flat CCN spectra and low k values, while hydrophobic and ultrafine (UF) mode (<100 nm) AP will have steep CCN spectra and high k values (Hegg et al., 1991; Jefferson, 2010) as those particles need





higher SS to activate as CCN. Thus, the empirical fit parameter 'k' indicates the nature of the aerosol system towards CCN
activation and 'C' indicates the CCN concentrations at 1.0 % SS. High C and k values are reported for the anthropogenic,
while low values are reported for the natural/marine AP (Seinfeld and Pandis, 2016; Andreae, 2009). From Figure 4, it can be
seen that the mean k values are higher during the monsoon conditions than the continental conditions. The highest k value
(~0.67) is observed during August and the minimum (~0.52) during June. As may be noted,  bigger or hygroscopic (or both)
particles which are CCN active were abundant during June compared to the monsoon months, when fine or hydrophobic (or
both) particles were predominant.

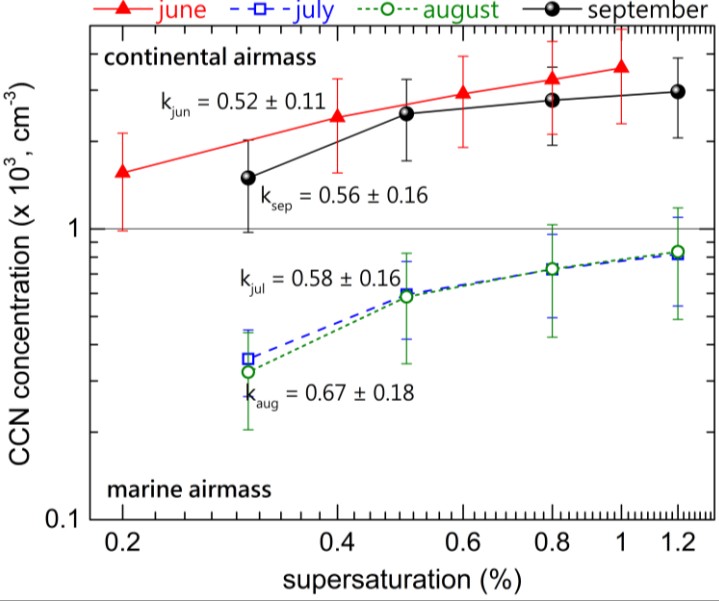

**Figure 4.** Mean CCN concentrations (± standard deviation) for different SS during continental (June and September) and marine (July and August) conditions. The power-law fit k-value of each spectrum is also given.

255          CCN concentrations and the k values reported during the current study, along with a few other studies are given in
Table 2. Generally, most of the aerosol abundance measurements such as BC mass (Kompalli et al., 2014), and aerosol number
concentration (Babu et al., 2016) showed the lowest seasonal mean value during ISM over the Indian region, mainly due to
the wet removal of AP. The CCN concentration at semi-arid Solapur before the onset of ISM is comparable to the values
(~2,000 cm⁻³) observed over the arid north-west region of India reported by Jayachandran et al., (2019). Interestingly, the CCN
concentrations at Solapur during active monsoon period is the lowest among the values reported over the Indian sub-continent.
The current values (~350 cm⁻³) are comparable to those reported from Ponmudi (~400 cm⁻³) at the southern part of the WG,
and another site at WG - Mahabaleshwar (~500 cm⁻³) at 0.2 % SS. From Table 2, very high values of CCN concentrations are
reported from polluted urban environments. Very low CCN concentrations (<300 cm⁻³ at 1% SS) are also reported from pristine
environments like Amazon (Pohlker et al., 2016; 2018), and Alps (Juranyi et al., 2011). The mean CCN values observed at
Solapur during ISM are comparable to those classified as 'polluted-marine' by Andreae, (2009).





An enhancement in k-values are observed during the monsoon period. Jayachandran et al., (2017; 2018) reported similar results for the monsoon period both at a coastal site, and at a hill station in the WG. From the southern tip of India, Jayachandran et al., (2017) have shown that the enhancement of k values (~0.7) associated with wet scavenging and lower k values (~0.55) during no rainfall conditions, within the same ISM period. The enhancement in k values (two-times) associated with the monsoon rainfall can be seen from Mahabaleshwar also (Table 2). Thus, the CCN concentrations at different SS at Solapur during ISM are low compared to those reported from other environments in India, while the CCN spectra show the common characteristics to those values reported from WG and peninsular India.

Significant diurnal variations are seen in the PBL AP properties over the Indian sub-continent (Nair et al., 2007). Daytime high and nighttime low aerosol abundance characterized by anomalous high values just after sunrise is known as the fumigation peak (Prakash et al., 1992), is generally observed. This diurnal pattern is mostly due to the evolution of the PBL and due to local emissions (Nair et al., 2007; Kumar et al., 2015). As the CCN activation and its properties are highly heterogeneous, it is very important to know its variation in a day. The diurnal variation of CCN concentration at 0.3 % SS, segregated to wet and dry conditions are shown in Figure 5(a) and 5(b), respectively. The CCN variations in a day are similar during the wet months (Fig. 5a), while it differs before and after the monsoon rainfall. In general, CCN concentrations show a slight enhancement (more prominent during clean background-marine air mass) during daytime due to the anthropogenic activities. A rapid increase is seen just after the sunrise in all months, except September, which is due to the mixing of the nocturnal residual layer with the evolving PBL (fumigation peak). There is no vivid diurnal variation in CCN during September. The diurnal variations of CN and BC concentrations for different meteorological conditions are shown in Figure 6. The diurnal variation of the total AP concentration and BC mass concentration is more vivid than that of the CCN concentration. A clear bi-modal variation is seen in both CN and BC diurnal variations during July an August. A sharp peak is seen in both CN and BC after the sunrise (0600-0700 IST) and the next peak starts increasing from 1500 IST maximum is at around 2000 IST. The diurnal variations in CN and BC are less prominent during the wet months, compared to that during the dry months. In both, the conditions a small increase is seen in the CN concentration around mid-noon. The fumigation peak seen in BC in the early monring was of the dry conditions such and were more than twice the daytime average values. Another important observation is that the CN values were consistently higher during September throughout the day than June, while the daytime BC mass was higher during June than September. In contrast to the CCN, the BC had well defined multiple peaks (morning and evening), indicating the contrasting aerosol source characteristics during the diurnal cycle. The well-mixed conditions are reached during the late afternoon hours and the PBL mixing has a well defined role in the reduction of concentration during the daytime, until new sources of aerosol are injected in to the atmosphere in the evening hours. The nighttime increase in BC could be attributing the stable conditions and less vertical mixing. The role of the PBL in modulating regional aerosol characteristics will be dealt in a separate study.





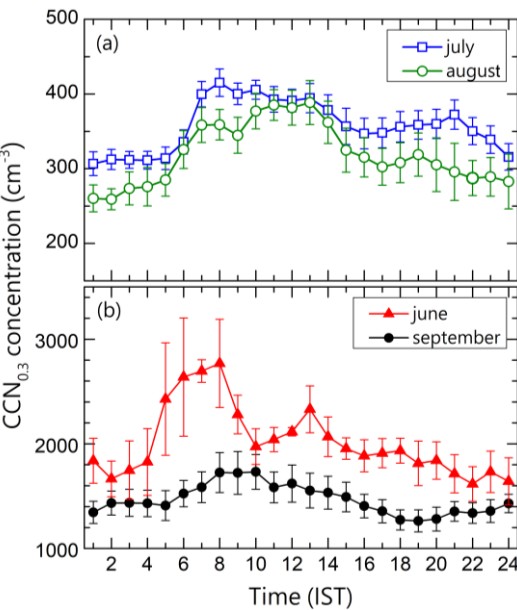

**Figure 5.** Diurnal variation of CCN concentration at 0.3 % SS during (a) wet and (b) dry conditions.

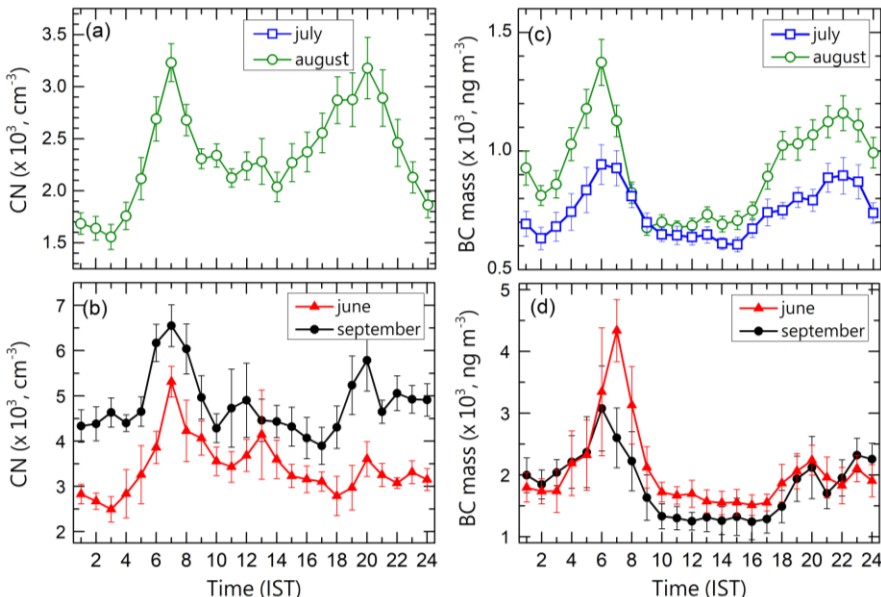

**Figure 6.** Diurnal variations of CN during (a) wet and (b) dry conditions, and BC mass concentrations during (c) wet and (d) dry conditions.

A few studies report the diurnal variations in CCN and its properties; at coastal (Jayachandran et al., 2017), Western Ghats (Jayachandran et al., 2018; Leena et al., 2016), rain-shadow (Varghese et al., 2016), IGP (Patidar et al., 2012), central Himalayan (Gogoi et al., 2015), and Eastern Himalayan (Roy et al., 2017) environments of the Indian sub-continent. Weak

diurnal variations in CCN concentrations during ISM, similar to the present study but with opposite patterns were reported





from the southern coast by Jayachandran et al., (2017) and from WG by Leena et al., (2016). Day-night variations in the CCN concentration can be due to the changes in aerosol sources, PBL dynamics, or both. Since the sky is generally overcast during the ISM and hence a shallow moist PBL (Sandeep et al., 2014) prevails, the observed diurnal variations are mainly due to the diurnal variations in the source and sink processes. The bi-modal diurnal pattern seen in BC mass concentration at Solapur is

seen similar to the observations reported by Safai et al., (2007) over Pune. Apart from the fumigation process happening during the sunrise, vehicular and biomass emissions also have a role in the peaks observed in a day. Thus, both local emissions and PBL dynamics contributed to the diurnal variations observed in the AP characteristics. The less diurnal variations during the dry conditions indicate the consistently high AP background conditions over Solapur, while the evident diurnal variations during the wet conditions indicates the presence of significant local aerosol sources

**3.3 CCN-CN association**

The association of CCN concentration at 0.3 % SS with the concurrent total AP number concentration, CN (~15-685 nm) is investigated separately for different conditions and is shown in Figure 7. CCN concentration at 0.3 % for the month of June is estimated from the measured CCN spectra. Though CCN forms the sub-set of the total AP system, the highly uncertain and non-linear dependence of CCN on the total AP is worth investigating. The role of the aerosol NSD is revealed through the

color of the scatter which represents the Geometrical Mean Diameter (GMD) of the corresponding AP system. It can be inferred from the Figure that the relationship between CCN and CN is different for different conditions. A least-square linear fit forced through origin (as there is no CCN in the absence of CN) is made through the scatter and the corresponding fit parameters and the linear fit line is also shown in the Figure.

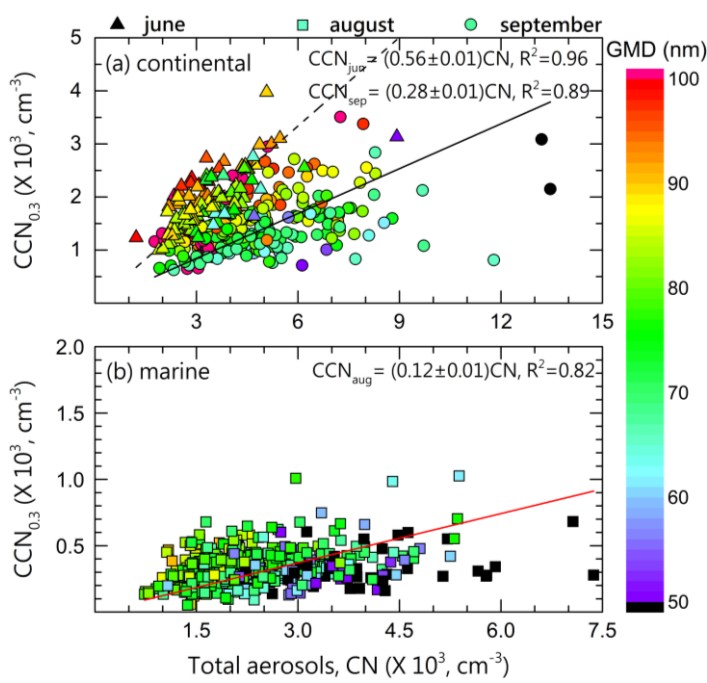



**Figure 7.** Association between CCN (at 0.3 % SS) and CN concentrations segregated to (a) continental and (b) marine conditions. The color of the scatter indicates the concurrent geometrical mean diameter of the aerosol system. The least-square linear fit is also shown along with the fit parameters.

The best linear relationship between the CCN at 0.3 % and CN concentrations is seen during June, and the
corresponding slope and correlation coefficient (R) of the fit are ~0.56 and 0.98, respectively. The linear association weakens during the monsoon condition when the slope and correlation coefficient values of the fit reduce to ~0.12 and 0.90, respectively. Even though under continental conditions, the slope of the linear fit during September (~0.28) reduces to the half of that measured during June, and the correlation coefficient value (R=0.94) also reduces. The relationship between CCN and CN is weak during August and only a few AP are activating as CCN. It can be seen that most of the scatter points which lies below
the linear fit line and corresponding to the higher (than the monthly mean) CN values are having GMD less than 50 nm. Even though the number of cases is less, similar observations can be seen during the continental case also. The two points (black-circle) corresponding to CN concentrations higher than 13,000 cm$^{-3}$ are having GMD less than 50 nm. These cases which reduce the CCN activation indicate the presence of an UF mode, probably due to the new particle formation (NPF) events. However, the presence of UF particles is not the only cause for less activation of CN as CCN in monsoon months as the scatter
and the linear fit excluding the UF particles are also having low correlation and slope values.

During August (Fig. 7b), CCN concentration at 0.3 % seems to be not varying linearly with the CN and is nearly constant at ~600 cm$^{-3}$, despite of CN concentration increasing to ~7,500 cm$^{-3}$. This is indicative of a significant number of UF or Aitken mode particles that require high SS for activation. Similar behavior of AP system towards CCN activation is observed at Eastern Himalayas (Roy et al., 2017). The drastic difference in CCN-CN association, similar to the present study, is also
reported by Asmi et al., (2012) between winter and summer months at a high-altitude site in France. They have attributed the predominance of accumulation mode particles and fine mode particles during winter and summer months, respectively. From the Central Himalayas, Dumka et al., (2015) have shown the increase in scatter during ISM due to the change in the aerosol physico-chemical properties. The spread of the scatter between CN and CCN increases for polluted conditions (Jayachandran et al., 2019), which is mainly due to the complex aerosol size distribution and mixing state. Thus, the CCN dependence on CN
population during the ISM shows a complex dependence on the aerosol size and mixing state.

### 3.4 CCN Activation Fraction

The fraction of AP acting as CCN at a given SS is known as the CCN activation fraction/ratio (AF) and is an important parameter to characterize the CCN activity (Dusek et al., 2006; Andreae, 2009; Deng et al., 2013). The CCN AF values for all the SS segregated to different air mass history is shown in Figure 8. During June, more than 40 % of the AP are getting
activated as CCN at 0.2 % SS. Nevertheless, during August, about 40 % of the particles only are activating as CCN even at 1.2 % SS, revealing the highly CCN in-active nature of the aerosols. During September, the CCN AF values at all SS are





between those of June and August. The CCN AF values at all SS show a similar feature in CCN-CN relationship at 0.3 % SS (Fig. 7).

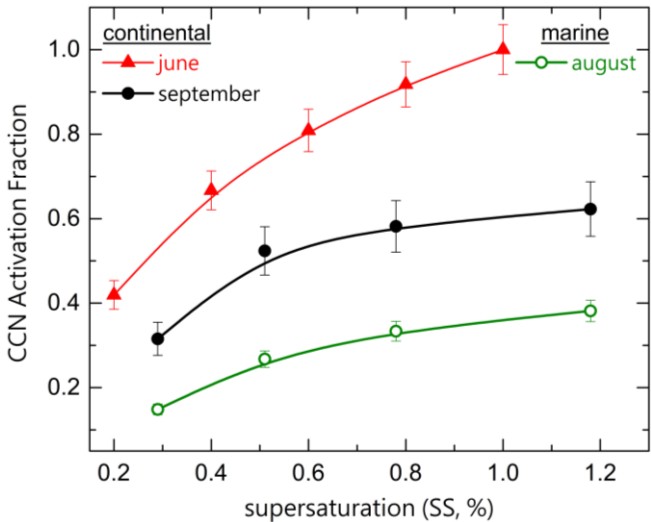

**Figure 8.** Variation of CCN activation fraction at different SSs during continental (June and September) and marine (August) conditions. Vertical error bars indicate the standard error.

The diurnal variations of k and AF (0.3 % SS) values segregated for the wet and dry conditions are shown in Figure 9. Unlike the dry conditions, the k values show a clear diurnal pattern during the wet conditions, similar as in the case of CN and BC concentrations. The k values increase almost twice after the sunrise and decrease thereafter reaching the nighttime values by evening hours (1600 IST) during July and August. Again, the k values peak at around 2100 hours IST. The enhancement in CCN and k values during daytime in marine airmass conditions, indicate the influence of local anthropogenic aerosol sources in determining the CCN activation. As discussed in Figure 8, the CCN AF values are very low throughout the day, with a slight increase during noon hours in August, when the k-values are low. Contrastingly the AF values are consistently high throughout the day during June. During September, an increase in CCN AF (from ~0.3 to 0.4) can be seen during the daytime.

It is well understood that the CCN characteristics are a function of aerosol size and composition. Hence it will be interesting to know the diurnal variations of the concurrent aerosol size and composition. In the absence of continuous aerosol composition measurements, the absorption Angstrom exponent ($\alpha_{abs}$), which is a proxy to identify the nature of the carbonaceous aerosols is estimated, and its diurnal variation for different conditions are shown in Figure 10 (a and b). The diurnal variation of the GMD for the corresponding periods are shown in Figure 10 (c and d). The diurnal variations of $\alpha_{abs}$ is similar for July and August months with values peaking by sunrise (0600-0900 IST) and late evening hours (1800-2000 IST). Almost the same pattern is seen during September month also, but of different magnitudes. Meanwhile there is no clear diurnal variations in $\alpha_{abs}$ during the June observations similar to the diurnal variations of 'k' and AF.






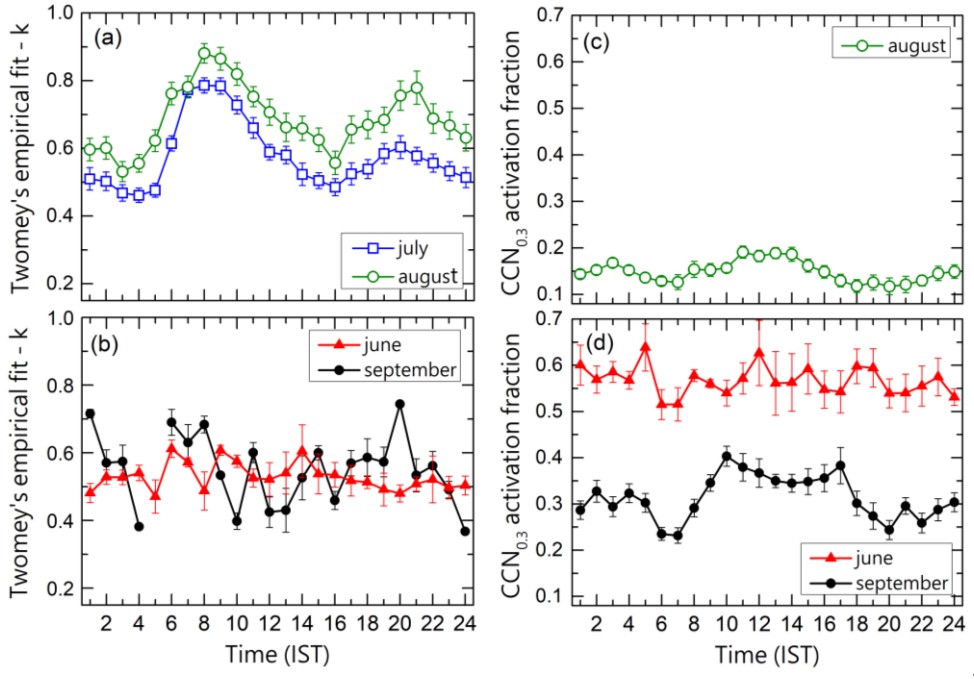

**Figure 9.** Diurnal variations of k-values (a and b), and CCN activation fraction (c and d), during marine and continental conditions. Vertical bars indicate the standard error values.

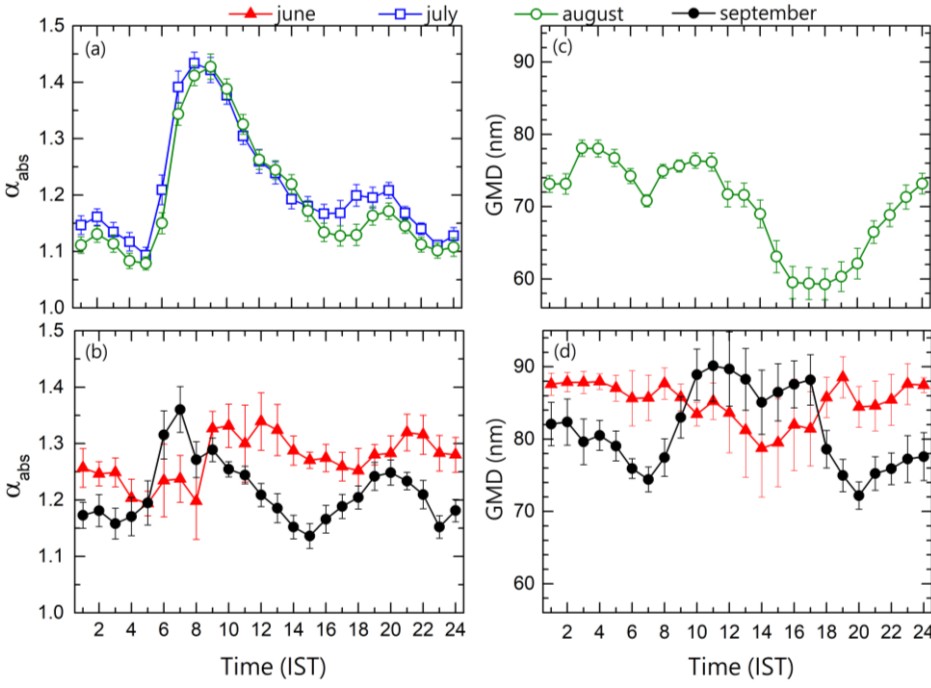

**Figure 10.** Diurnal variation of absorption Angstrom exponent (a and b) and geometrical mean diameter (c and d) during continental and

marine conditions. Error bars indicate the standard error associated with the mean values.





The mean GMD values during the marine conditions decrease from 1200 IST onwards and reaches the minimum value (<60 nm) from 1600 to 1800 IST, which increases back to ~72 nm by midnight hours. Interestingly, a small dip in the GMD is observed during the fumigation peak (~0700 IST), associated with the sharp increase in $\alpha_{abs}$ values. The GMD values
were consistently higher throughout the day during June, such that the lowest mean value (~79 nm) observed at the 1400 IST is higher than the maximum mean GMD (~78 nm) observed in a day (3 and 4 IST) during August. GMD during September depicts a clear diurnal variation which is opposite to that observed during June, with distinct high values during daytime. Similar to August, GMD decreases during the fumigation peak associated with the sharp increase in the $\alpha_{abs}$ values.

Even though the mean $\alpha_{abs}$ values are almost similar in all the months with comparatively higher (1.29 ± 0.09) during
June and lower during the August (1.19 ± 0.14), the values show diurnal variations systematic with aerosol abundance diurnal variations. The sudden sharp increase in the k values (Fig. 9a) during sunrise hours of August is associated with a similar enhancement in the $\alpha_{abs}$ values. The high k values (>0.8) during these hours is due to the organic aerosols, inferred from $\alpha_{abs}$ values. Chung et al., (2012) have reported $\alpha_{abs}$ values above 1.6 for organic aerosols while, Gyawali et al., (2009) have reported $\alpha_{abs}$ values above 1.4 for biomass smoke. The daytime enhancement (~2 times) in CCN AF during September is
exactly according to the daytime enhancement seen in the aerosol GMD. Jayachandran et al., (2017) have reported the similar association between CCN AF and aerosol GMD diurnal pattern during ISM from a coastal site in southern peninsular India. Interestingly, similar association is not seen in other months. Thus, the aerosol composition especially the organic aerosols inferred from the high $\alpha_{abs}$ values is playing a major role in determining the CCN activation during the monsoon conditions, while the aerosol size is determining the CCN activation during the continental conditions.

In general, high AF is found for aged background aerosols, while freshly emitted polluted urban aerosols have low CCN efficiency (Andreae and Rosenfeld, 2008). CCN AF values reported from India and some relevant studies reported across the globe are mentioned in Table 2. The similarity seen in CCN concentration and k values are seen in CCN AF also between Solapur and Ponmudi during the ISM. At both the places, only a small fraction (15-20 %) of the ambient AP is activating as CCN at 0.3 % SS. As seen in the Table, high AF values are reported from the coastal location and central Himalayas. The high
CCN AF during the continental conditions at Solapur is similar to those reported during dry conditions in Nainital (Gogoi et al., 2015), possibly associated with biomass burning. The low CCN AF observed at Solapur during monsoon rainfall, possibly resulting from the wet scavenging, is consistent with the values reported over the sub-continent during similar conditions, while the high CCN AF before and after monsoon rainfall is observed by several studies, resembling a biomass burning dominant polluted environment (Andreae, 2009).

As mentioned earlier and reported by several studies (Dusek et al., 2006; McFiggans et al., 2006), aerosol size plays a major role in determining the CCN activation ability of aerosols. It has been found that the UF particles were present during the monsoon conditions, when CCN AF was very low (Fig. 8). Meanwhile, the presence of bigger particles is enhancing the CCN activation in other cases. To investigate the role of aerosol size in the observed CCN activity, the aerosol NSD during each condition is examined in detail.





## 3.5 Aerosol size distribution and critical activation diameter

The simultaneous and co-located aerosol size distribution observations and critical diameter are examined. The fraction of particles in the nucleation mode, Aitken mode, and accumulation mode is estimated and shown (in %) in Figure 11(a). Nucleation mode particles are those observed below 30 nm, Aitken mode particles are those from 30 nm to 100 nm and accumulation mode particles are those beyond 100 nm. The corresponding mean NSD of AP along with the standard deviations for the study period are shown in panels (b), (c), (d) of Figure 11. The frequency of occurrence of the GMD for each month along with the mean GMD values are shown in Figure 11(e). As seen in the CCN characteristics, aerosol NSD also depicts distinct features prior to the monsoon (Fig. 11b), during monsoon (Fig. 11c) and after the monsoon (Fig. 11d).

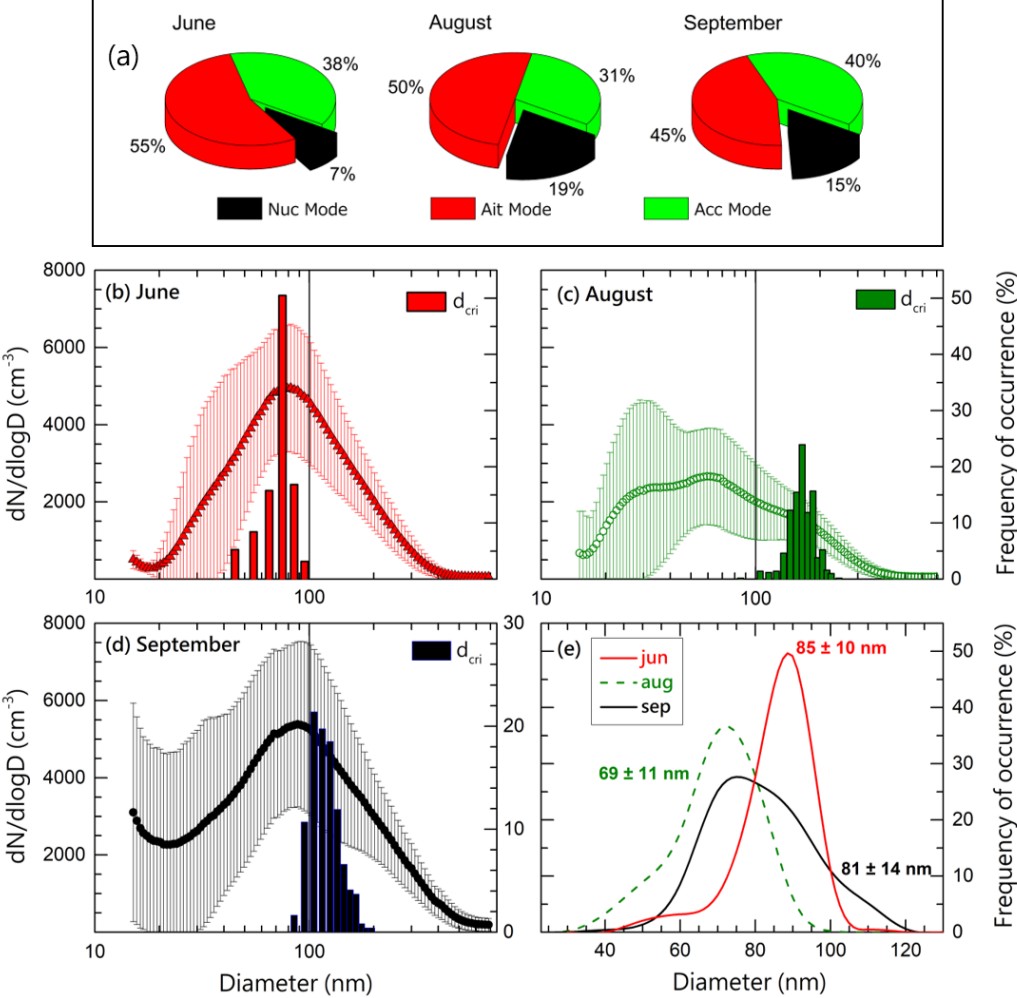

**Figure 11.** (a) Fraction (in %) of nucleation mode, Aitken mode, and accumulation mode particles during the observation periods. Aerosol mean number-size distribution (with standard deviations) during (b)June, (c) August, and (d) September months. The bars in the same plots indicate the frequency distribution of critical activation diameters at 0.3 % SS of the corresponding months (right axis). (e) Frequency distribution of the geometric mean diameter of the aerosol system during the observation periods.



During June, most of the distributions are mono-modal, peaking around 80 nm and the mean GMD during this period is ~85 nm. In this period, the majority (~55 %) of the AP were present in the Aitken mode and least (~7 %) in the nucleation mode. During similar continental air mass conditions, but after the monsoon rainfall, the aerosol size distributions during the September is also similar. But a prominent presence of nucleation mode particles (15 %) is also seen in the Figure 11(d). The consistent presence of such particles is seen as the spread of the distribution of the GMD in Figure 11(e). Unlike the continental air mass conditions, the aerosol size distributions are entirely different during the marine air mass under the monsoon rainfall conditions as seen in Figure 11(c). Three modes are distinctly observed in the mean picture, with two peaks below 100 nm. The mean GMD during this period is ~69 nm with the frequency distribution spreading towards the lower size range. About 19 % of the total aerosols were found in the nucleation mode (<30 nm) during August and this feature continues in September also, even though the air mass history changes. The accumulation mode (30-100 nm) AP concentration diminished (only 31 %) during the August.

For a given aerosol NSD, the critical activation diameter ($d_{cri}$) serves as an important parameter for representing the CCN activity, along with the CCN activation fraction and the empirical fit parameter – k values. Assuming homogeneous composition, $d_{cri}$ for a specific SS can be estimated by integrating the aerosol NSD from the higher to lower size, until the integration becomes equal to the measured CCN number concentration at that SS (Furutani et al., 2008; Kammermann et al., 2010; Deng et al., 2011; Varghese et al., 2016; Fang et al., 2016). The lower limit of the integration can be considered as the 'apparent' critical activation diameter, as the ambient aerosol system can have both internal and external mixing state, and size-dependent composition. The frequency distribution of $d_{cri}$ estimated for 0.3 % SS is also shown as the bars (right axis) in Figure 11(a-c). Comparing the $d_{cri}$ for different conditions, the values were always less than 100 nm during June, contrastingly the values were always greater than 100 nm during August. The $d_{cri}$ values were around 100 nm during the September month. The mean (± standard deviation) values of $d_{cri}$ were 72 ± 12 nm, 169 ± 38 and 121 ± 20 for June, August and September months, respectively.

Different factors such as heterogeneous sources (Kim et al., 2002; Morawska, 2002), local meteorology (Wehner and Wiedensohler, 2003; Du et al., 2018), long-range transport (Birmili et al., 2001), and cloud processing (Noble and Hudson, 2019) can influence and modify the NSD. The predominant (60 %) fine particles (<100 nm) size distribution (bi-modal) during monsoon is similar to the two modes observed in the fine size range during monsoon at the urban site, Kanpur (Bhattu and Tripathy, 2014) during the ISM. The GMD values and the corresponding CCN properties from the present study and relevant other studies are listed in Table 2. Less AP concentration with low GMD (74 nm) was observed during monsoon at an urban site by Kanawade et al., (2014) and at a background Himalayan site (86 nm) by Kompulla et al., (2009) over the Indian sub-continent. Similar to the present study, GMD was higher before and after the monsoon period, in both studies. From a high-altitude site in WG, Leena et al., (2016) have reported the lowest seasonal mean GMD of ~77 nm during ISM. The enhancement of the smaller particles in the total aerosol system, causing the reduction of the GMD especially during monsoon as seen in the present study is consistent with the previous studies (Babu et al., 2016). The mean GMD value observed over Solapur during ISM is the lowest (69 nm) reported value during the similar period over the Indian region.



Similar to the present observations, comparable accumulation and Aitken modes, and a dominant accumulation mode was reported over Amazon for wet and dry months, respectively by Pohlker et al., (2016). The accumulation mode particles are usually aged, associated with biomass burning (Kalvitis et al., 2015), and mostly resulting from the condensation of

secondary inorganics and organics, and coagulation of smaller particles (Seinfeld and Pandis, 2016). Interestingly, smaller aerosol GMD values (<70 nm), similar to the present observations during August, were consistently observed near anthropogenic sources by Quinn et al., (2008). In the same study, they found bigger AP (GMD>70 nm) for observations carried out away from anthropogenic sources, which is similar to the present observations during June and September. From the concurrent Aerosol Mass Spectrometer measurements, they found that hydrocarbon-like organic aerosols (HOA) having mass

spectrum characteristic of long chain hydrocarbons from fresh diesel exhausts were responsible for the fine mode and oxygenated organic aerosols and sulfates are responsible for the higher GMD. Hence the presence of freshly produced local fossil fuel combustion aerosols in the UF mode can be the reason for the low CCN activity during August in the present study. The low CCN activity in August could also be due to the wet scavenging conditions that were prevailing over the site, which indeed contributes to significant washout of bigger and hydrophilic AP.

Since the CCN activity depends mainly on the aerosol size and chemical composition (Dusek et al., 2006; McFiggans et al., 2006), $d_{cri}$ estimated from concurrent aerosol NSD and CCN measurements can be considered as a proxy for the variations in the chemical composition of the aerosol system. As the aerosol size distribution and chemical composition are intrinsically associated with each other, any shift in the physical size distribution is mostly associated with the change in the aerosol composition, arising mainly due to the change in the sources or due to different processes such as ageing, coating or

scavenging, except for externally mixed systems (Crosbie et al., 2015). Quinn et al., (2008) have correlated the $d_{cri}$ with the HOA mass, and found that HOA can explain about 40 % of the variance in the $d_{cri}$. They have reported 70-90 nm and higher values (>90 nm) as the $d_{cri}$ for marine and inland regions, respectively, at 0.44 % SS. For anthropogenic and marine environments, Furutani et al., (2008) have reported $d_{cri}$ values of 70-110 nm and 50-60 nm, respectively, at 0.6 % SS. The $d_{cri}$ values observed during continental conditions at Solapur is similar to the values observed at a tropical monsoon climate region

by Fang et al., (2016), under urban influence. The $d_{cri}$ values during August are higher than the corresponding values reported from polluted North-China plain by Deng et al., (2011). From an urban site, Burkart et al., (2011) have reported an average value of ~169 nm for $d_{cri}$ at 0.5 % SS. Freshly emitted carbonaceous combustion particles can have large $d_{cri}$ values up to ~350 nm, even at a high SS (0.7 %) (Hitzenberger et al., 2003). The $d_{cri}$ can exhibit very low values also, than theoretically estimated ones in the presence of partially or fully soluble particles, as their slight presence can largely enhance the CCN

activity of insoluble particles such as BC and dust (Dusek et al., 2006; Begue et al., 2015). The sharp distinction in the $d_{cri}$ values and aerosol NSD between different atmospheric/air mass conditions within the same season in the present study indicate the difference in the aerosol composition. The low GMD and high $d_{cri}$ during the August indicate the presence of freshly emitted water-inactive primary organic aerosols.

**Relationship between CCN activation fraction and critical activation diameter**





500       The association of CCN activation fraction with $d_{cri}$ is examined and shown in Figure 12. The different months are indicated by the shape of the scatter, while the color indicates the concurrent UF fraction (CN<30 nm in %). In general, the CCN AF and $d_{cri}$ are anti-correlated. From the Figure, it can be seen that the points corresponding to June (triangles) and August (stars) months lie at opposite corners of the scatter, while those during September (circles) are between the other two. As seen in the aerosol NSD discussions, nucleation mode AP are almost absent during June, while it is more common during

August and a few cases are observed during September. But there are cases of having low CCN AF (<0.1), but not associated with UF events, but associated with higher BC mass concentrations. Thus, the presence of UF particles cannot be pointed out as the only cause for the lower CCN efficiency of the aerosols, but the composition is also having a role in determining the CCN efficiency.

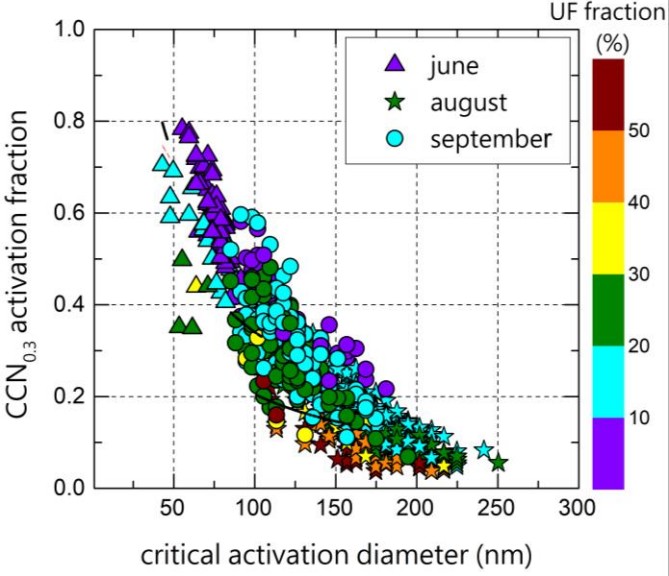

**Figure 12.** Scatter between CCN activation fraction at 0.3 % SS and the critical activation diameter. The color of the scatter indicates the nucleation mode (UF) fraction.

      From Figure 12, it can be seen that the CCN AF is not varying linearly with the $d_{cri}$. Except for the month of June, the CCN AF exhibited a power law dependence on $d_{cri}$. Hence a non-linear fit was applied to the scatter as

$$CCN_{AF} = a \times d_{cri}^{-b} \qquad (3)$$

'a' is a constant as the CCN AF is assumed to be one when the $d_{cri}$ reaches zero, in all the cases. The value of 'b' and the corresponding correlation coefficient (inside bracket) of the fit was estimated as 0.67 (0.97), 0.73 (0.97), and 0.84 (0.98) for June, September, and August months, respectively. During June, the CCN AF (in %) decreases linearly with $d_{cri}$ with a negative slope value of 0.59 with a better correlation coefficient of 0.99, than that obtained using the non-linear fit.

520       The inverse-linear relationship between the CCN AF and $d_{cri}$ during June is similar to that reported by Furutani et al., (2008). In their study the critical diameter varied within a range (30-130 nm) similar to that of the present observation





during June (40-90 nm), where the linear relationship exists. As the $d_{cri}$ increases beyond 100 nm, linear relationship switches to non-linear, and the order of the relationship increases as $d_{cri}$ increases, which is illustrated by September (b = 0.73) and August (b = 0.84) months values.

The aerosol composition plays an important role associated with the changes in the aerosol NSD due to the meteorological processes and active source and sink mechanisms prevailing during the monsoon conditions. The significant influence of aerosol composition in determining the CCN activity at lower SS which is more probable in real atmosphere, is demonstrated in many studies (Cubison et al., 2008; Kammermann et al., 2010; Bhattu et al., 2015; Jayachandran et al., 2017). In this aspect, the role of carbonaceous aerosols in determining the CCN activation is investigated.

The association of CCN concentration at 0.3 % SS and absorption coefficient at 550 nm, segregated for different periods are shown in Figure 13. Each point in the Figure indicates the CCN at 0.3 % SS corresponding to the aerosol absorption coefficient given in the X-axis. The color of the scatter indicates the $\alpha_{abs}$ values. A least square linear fit of the scatter is also made and shown along with the fit parameters. The association between aerosol absorption and CCN concentration is generally low, but comparatively high during the continental air mass and very weak during the monsoon conditions. The better

association between the CCN concentration and absorption properties may be due to (i) absorbing AP itself acting as CCN or (ii) aerosol species co-emitted with the absorbing AP activating as CCN. The higher slope and better association observed during June indicate that the low k, high AF values and the high association of between CCN and CN during the period are due to the major contribution of carbonaceous aerosols towards CCN activation. It can be either due to the co-emitted organics enhancing the CCN efficiency of the aerosol system or due to the aged carbonaceous aerosols itself activating as CCN or due

to the combination of both. The enhancement in the accumulation mode aerosols supports this observation as oxygenated organic aerosols and sulfates are found in the accumulation mode (O'Dowd et al., 1997; Quinn et al., 2008). During marine air mass (wet conditions) there is no clear association between aerosol absorption and CCN, and less AP are activated as CCN. It indicates that there is a change in the source/sink and nature of CCN during marine/wet and Continental/dry conditions.

Carbonaceous aerosols form a major source of CCN concentration and thereby contribute to indirect effect of aerosols

(Novakov and Penner, 1993). Anthropogenic carbonaceous aerosols cause an indirect effect of -0.9 W m$^{-2}$, while sulfates cause only -0.4 W m$^{-2}$ (Lohmann et al., 2000). Spracklen et al., (2011) have shown through simulations that about 60 % of the global CCN concentration is from carbonaceous sources. Various atmospheric processes such as ageing, coating and mixing can enhance the water activity of BC (Lammel and Novakov, 1995; Kuwata et al., 2009), which is hydrophobic when freshly emitted.  Mixing with hydrophilic substances like inorganic salts can also enhance the CCN activity of carbonaceous aerosols

(Dusek et al., 2006). Thus, the better correlation observed between CCN and aerosol absorption, and the associated high CCN efficiency during the continental conditions indicate the significant role of carbonaceous aerosols in CCN activation at Solapur. Jayachandran et al., (2018) illustrates the close association of CCN with aerosol absorption properties (better than present study) from WG and the lack of association between the parameters at a coastal site during the monsoon conditions. Enhancement in CCN concentration along with increase in the aerosol absorption coefficient was observed at central



Himalayas (Gogoi et al., 2015). In general, carbonaceous aerosols have a significant role in CCN concentration during continental conditions.

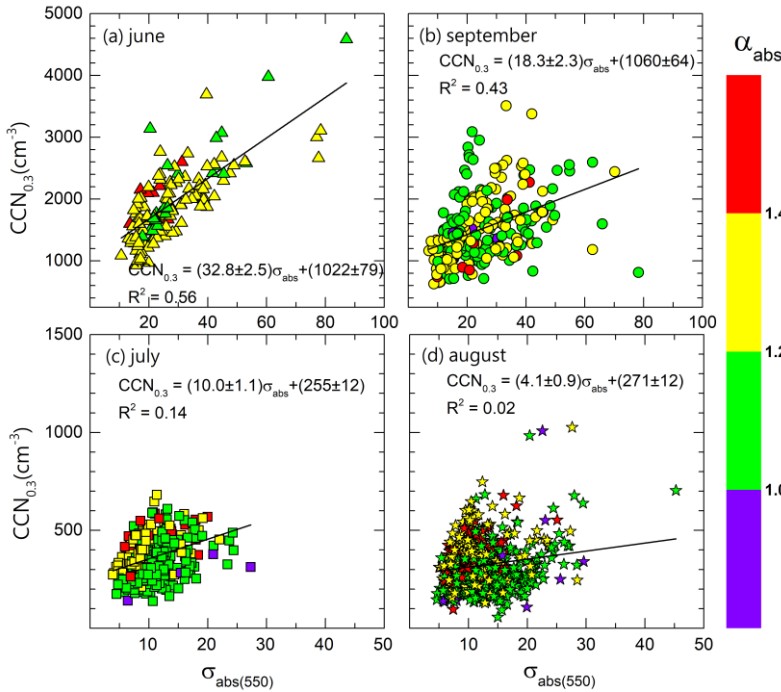

**Figure 13.** Association between CCN (at 0.3 % SS) and absorption coefficient (at 550 nm) for (a) June, (b) July, (c) August, and (d) September months. The color of the scatter indicates the concurrent absorption angstrom exponent values. The least square linear fit is also
shown along with the fit parameters.

As the very low AF is observed along with the enhancement in the nucleation mode particles and depletion of Aitken particles after the onset of ISM over the study region, the chemical characteristics of Aitken, as well as accumulation mode particles, are investigated. Most of the studies, from different parts of the globe, have reported about the high hygroscopicity
of accumulation mode particles and less hygroscopicity for the Aitken mode particles (Paramanov et al., 2013; 2015). From the long-term observations from Amazon, Pohlker et al., (2016) have concluded that organics predominantly present in the Aitken mode reduces the hygroscopicity, while the dominance of inorganics in the accumulation mode enhances the aerosol hygroscopicity which was in-line with other studies (Gunthe et al., 2009; C. Pohlker et al., 2012).

The CCN concentration found to have an association with the absorption coefficient during the continental air mass
compared with the marine air mass conditions. The reduced CCN efficiency due to the presence of Aitken or UF mode is already discussed. To ascertain the contribution of the carbonaceous AP to the NSD, the association of BC with Aitken mode particles and accumulation mode particles are examined and shown in Figure 14. A least square linear fit is also made and the corresponding parameters are shown in Table 3. The BC mass concentration is better associated with the accumulation mode AP during June and less associated during August. The BC mass concentration is not associated with the Aitken particles





during June, while comparatively better associated during August and September. The C values (Y-intercept) of the linear fit show that the BC mass is present even in the absence of Aitken particles in all the conditions and the highest during the June (C~1224 ng m$^{-3}$). Interestingly, the negative C values indicate, BC mass is absent while accumulation mode particles reduce to zero, except during monsoon conditions. This shows the BC or the co-emitted AP are present in the accumulation mode only during the continental (dry) air mass.

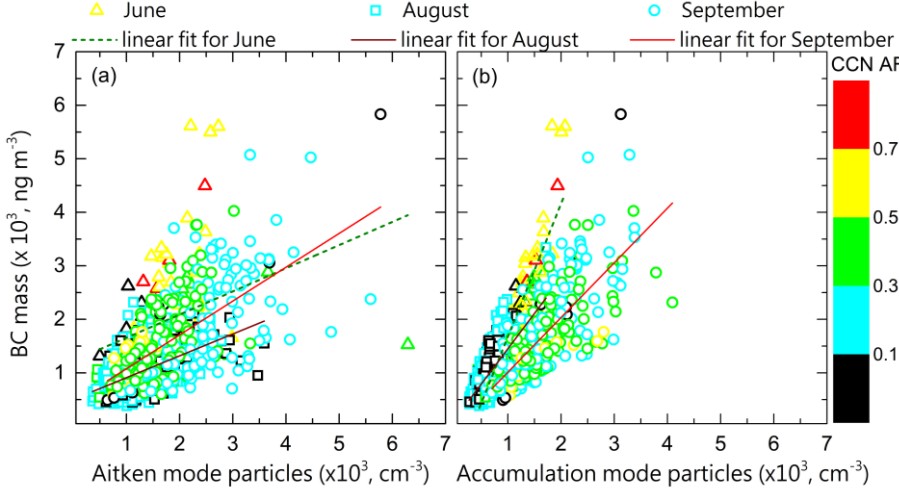


**Figure 14.** Association of BC mass concentration with (a) Aitken and (b) accumulation mode particles, segregated for different air mass conditions. The least square linear fit lines are also shown. The color of the scatter indicates the corresponding CCN AF values.

The association of BC with the accumulation mode particles during June along with high CCN AF indicates the aged,
large size hygroscopic particles from carbonaceous combustion sources present prior to the monsoon at the location. From the eastern coast of India, Kompalli et al., (2019) have reported highly coated larger BC particles (>110 nm) in dry conditions under the continental influence, while nascent BC particles (~80 nm) with less coating were found during ISM due to wet scavenging. This finding is in-line with the current observations. The enhanced fine particle concentration having better association with BC mass concentration after the onset of ISM underlines the possibility of freshly emitted carbonaceous
aerosols reducing the CCN AF. The association of BC mass and accumulation mode aerosol number concentration also points to the possibility of inorganic aerosols like sulfate, co-emitted along BC from carbonaceous combustion sources, enhancing the CCN activity of AP in dry, continental conditions.

**3.6 CCN closure**

The biggest uncertainty in ACI regarding the CCN concentration and characteristics is due to the lack of proper
understanding on the dependence of CCN on aerosol size and composition. To minimize this uncertainty, CCN closure studies have been carried out by many investigators (Brokehuizen et al., 2006; Lance et al., 2009; Juranyi et al., 2011; Bhattu et al.,





2015; Crosbie et al., 2015; Jayachandran et al., 2017) at different environments, leading to the better understanding on the CCN activation from aerosols. CCN concentration at the rain shadow region under different air mass conditions are estimated and validated with the measured CCN concentrations. The CCN concentrations are estimated by (i) assuming accumulation

mode aerosols activating as CCN (ii) applying the mean 'apparent' critical diameter and (iii) assuming the aerosol composition as ammonium sulfate ($(NH_4)_2SO_4$), and compared with observations. The scatter between the CCN concentration at 0.3 % SS estimated and the corresponding measured CCN concentrations segregated according to marine and continental conditions are shown in Figure 15. The color of the scatter plot represents the concurrent GMD values of the aerosol system. Least square linear fit is applied to the scatter and the corresponding fitting parameters are also mentioned in the Figure. The dash line

indicates the unit slope (m = 1) line.

CCN are generally approximated as AP above 100 nm (accumulation mode) in many studies when there are no concurrent CCN measurements (eg. Andreae, 2005). Still, it is a rough approximation due to the non-linear dependence of CCN activation of AP, and this assumption is examined in Figure 15 (a and b). The variations in accumulation mode aerosols are correlated well with the CCN concentration in continental conditions (R = 0.98 and 0.97), compred to the marine conditions

(R = 0.95). Interestingly under the same conditions (continental), the accumulation mode aerosols have different activation efficiency as CCN and is under-estimated (m = 0.64) during June and over-estimated (m = 1.26) during September. But during August the linear fit of the scatter indicates that the estimated CCN concentration is almost twice as that of the measured concentration, when the accumulation mode particles are considered to be CCN active.

Rather than taking an assumed value as the critical activation diameter (100 nm), the mean of the measured critical

diameter ($d_{cri}$) is used to estimate the CCN concentration and shown in panels (c) and (d) of Figure 15. The mean $d_{cri}$ for 0.3% SS of ~70, ~165 and ~120 are used for estimating CCN during June, August and September months, respectively. The estimated and the measured CCN concentration correlate well during the continental conditions with high correlation coefficient (R = 0.97) and almost unit slope. From the Figure 15(c), it can be seen that most of the points lie along the diagonal 1:1 line, irrespective of the number concentrations and GMD values. Compared to these, the CCN concentration is under-

estimated (m = 0.90) and correlation coefficient is less (R = 0.95) during August. The comparatively low correlation and slope of the linear fit indicate the absence of a sharp size cut in the activation of CCN. This may points to the role of chemical composition of small (<100 nm) particles and the mixing state of the aerosol system in CCN activation.

The CCN concentration at 0.3 % SS at different conditions are estimated (Fig. 15 e and f) by assuming an inorganic composition of $(NH_4)_2SO_4$. In all the conditions, CCN concentrations were over-estimated by this approach, indicating that

the ambient aerosol system has a lesser hygroscopicity ($\kappa$) than that of $(NH_4)_2SO_4$. The highest over-estimation (m = 3.38) is observed during August and the correlation of the linear fit of the scatter is also comparatively weak (R = 0.94). During June, the maximum correlation coefficient (R = 0.98) is obtained and the slope is also nearer to the unity (m = 1.13). However, even for the continental conditions, the CCN concentrations are over-estimated (m = 1.94) during September when $(NH_4)_2SO_4$ composition is assumed. These observations confirm that the nearly mono-modal aerosol NSD observed during June is more

similar to an aged continental aerosol system having similar hygroscopicity of sulfate aerosols. This observation has to be





considered along with the association of BC with accumulation mode aerosols (Fig. 14b). The AP system observed during marine conditions are of very less hygroscopic and the multiple size modes observed in the smaller size range indicate a heterogeneous composition in a complex mixing state, during the wet conditions.

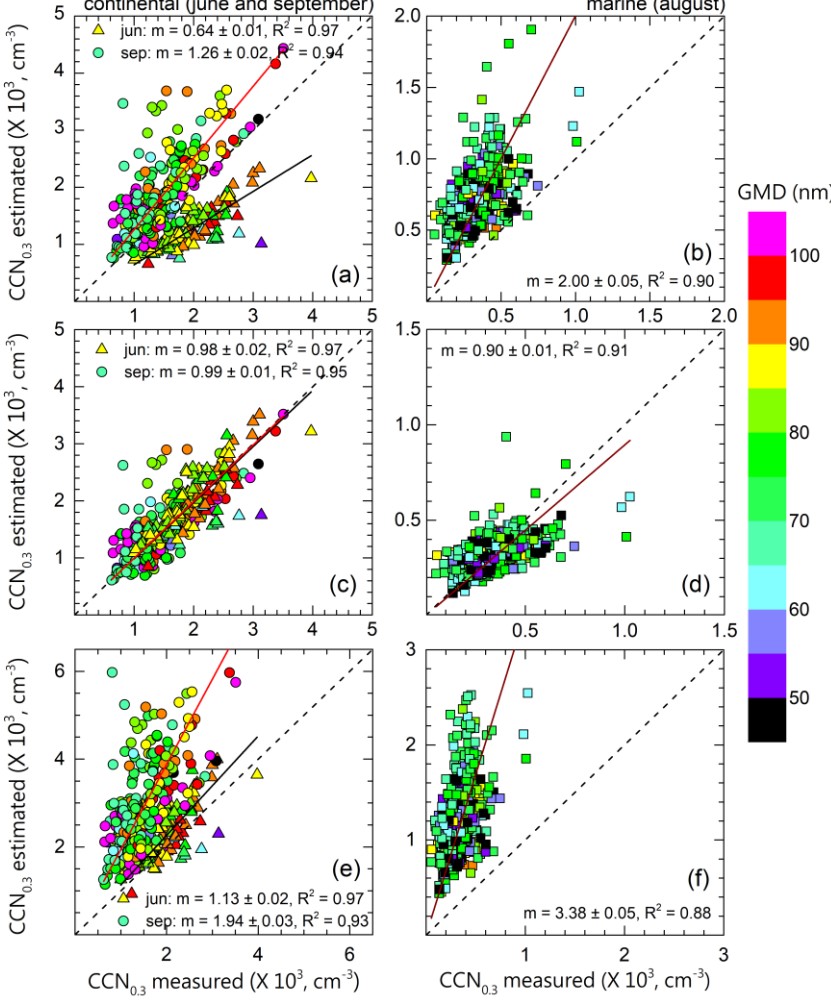

**Figure 15.** Scatter between estimated and measured CCN concentration at 0.3 % SS for continental and marine conditions. CCN estimated (a and b) as particles above 100 nm, (c and d) from critical activation diameter, and (e and f) using aerosol NSD and ammonium sulfate composition. Linear fit and the parameters are also shown. The dash lines indicate the unit slope (m=1) line.

Comparing the three approaches used to estimate the CCN concentration, using a sharp size cut for estimating CCN concentration suits well for the continental aerosol systems. Perfect closure is obtained using the measured critical diameter indicating the size dependency of CCN activation during these conditions. Still, there is a variation in the AP composition in the continental air mass before the monsoon rainfall and after that, reflecting in the different CCN efficiency of the accumulation mode during these periods. The aged aerosol system prior to the monsoon resembles a sulfate aerosol





composition with a very high CCN activation efficiency and low k values. The CCN AF during this period inversely varies in
a linear manner with the $d_{cri}$. During September all the accumulation mode aerosols are not participating in the CCN activation
(Fig. 15a) and the assumption of $(NH_4)_2SO_4$ composition nearly doubles the estimated CCN concentrations. Thus, fresh
anthropogenic fine aerosols mostly from carbonaceous combustion sources, during this period (Fig. 14a) are inactive towards
the CCN activation. Comparing the aerosol NSD in continental conditions, there is a depletion of Aitken mode particles and
an enhancement in nucleation mode particles during September. Similar to the different AF between June and September,
Pohlker et al., (2016) have showed high CCN AF in the absence of nucleation mode.

This study underlines the inability to make use of either aerosol size distribution or uniform (internally mixed)
composition to explain the CCN activation during the monsoon (wet) conditions. This finding is similar to the weak CCN
closure reported by Crosbie et al., (2015) for North-American monsoon conditions. The complex meteorological pattern
including the monsoon showers and regional aerosol production (both primary and secondary) causes large variability in the
aerosol NSD as seen in Figure 11. The very less hygroscopicity of the accumulation mode aerosols during the monsoon is
revealed in Figure 15(b). Even the measured mean $d_{cri}$ cannot predict the CCN concentration from the measured NSD
accurately. The least closure is obtained while assuming a uniform internal mixture of hygroscopic inorganic composition.
These all points to the highly complex mixture of the size-dependent composition of the prevailing aerosol system during
monsoon. Studies (Cubison et al., 2008; Ervens et al., 2010) have highlighted the need for size-resolved composition
information for estimating the CCN concentration for freshly emitted AP, near to the sources. It can also be noted that the
GMD may not have much role in deviating the closure during active monsoon, which is similar to the inference from Figure
7. Thus, even though the nucleation mode AP present during the period hinder the CCN activity, the presence of bigger particles
in the same period is not supporting the CCN activation. It indicates that apart from the size of the aerosols, the
composition/mixing state of the aerosol system during August is also playing a crucial role in determining the CCN efficiency.
From the aerosol optical properties (Fig. 10a), it is seen that the low CCN AF and high k value is associated with the
enhancement of the organics at the site. These organics observed after sunrise hours during monsoon conditions are limiting
the CCN activation of the aerosols. The quantification and classification of these species are essential to address the effect of
aerosols on clouds in a rain shadow region, especially during the monsoon conditions.

## 4 Summary and Conclusions

CCN characteristics at a rain-shadow region during the Indian Summer Monsoon (ISM) are studied with respect to
the different air mass and meteorological conditions that prevailed over the region. It is found that the polluted-continental
conditions transform into a polluted marine condition by the onset of ISM with a significant change in aerosol size distribution
and composition affecting the cloud nucleating properties. The important findings are listed below.

- Comparatively high BC (~2,000 ng m$^{-3}$) loading and AOD (> 0.5) prevailed over Solapur before and after the
monsoon, which reduced to very low values (BC~800 ng m$^{-3}$) during the monsoon-clean background conditions.



- The lowest CCN concentrations at any SS (~900 cm$^{-3}$ at 1.2 % SS) is observed at Solapur, compared to the values reported during ISM over the Indian sub-continent. However, the k values (~0.6) during ISM are high and similar to those reported over Western Ghats (WG) and peninsular India under similar conditions.

- A slight daytime enhancement in CCN is seen due to the influence of anthropogenic activities, while a significant enhancement in k values (2 times) was observed during the daytime of monsoon associated with organic aerosols, inferred from the concurrent high absorption Angstrom exponent values.

- Significant diurnal variations in CN, BC concentrations and properties like 'k', CCN AF and $\alpha_{abs}$ during the wet conditions indicate the dominant presence of local aerosol sources, while the weak diurnal variations of the same parameters during dry conditions indicate the consistent polluted background conditions at Solapur.

- The aerosol system prior to the onset of ISM having a mono-modal number-size distribution (NSD) with a geometrical mean diameter (GMD) of 85 ± 10 nm depicted high CCN activation fraction (AF) of ~55 % at 0.3 % SS. During the ISM, multiple modes were observed in aerosol NSD with a predominant nucleation mode fraction (~19 %) resulting in the lowest CCN AF of ~15 %. Just after the monsoon, aerosols were significantly present both in nucleation (~15 %) and accumulation mode (~40 %) and the CCN AF enhanced to ~32 % only even though the corresponding aerosols GMD was 81 ± 14 nm.

- The mean critical activation diameters ($d_{cri}$) estimated for 0.3 % SS from concurrent CCN and aerosol NSD measurements were highest during the monsoon (~165 nm) and lowest just prior to the monsoon (~70 nm), and ~120 nm just after the monsoon for 0.3 % SS. The CCN AF decreases with an increase in $d_{cri}$, linearly up to ~100 nm (prior to the monsoon) and beyond which non-linearly (during and after monsoon) decreases.

- Better association of absorbing type aerosols with CCN and accumulation mode aerosols during continental air mass conditions indicate the aged, bigger sized particles from carbonaceous combustion sources possibly enhancing the CCN activity prior and after the monsoon. CCN closure analysis revealed that the CCN population during continental air mass (before the monsoon) is more of sulfate type.

- The closure study indicates that the size dependency of CCN activation during dry-continental conditions weakens during wet-monsoon conditions. CCN estimation using measured aerosol NSD and sulfate composition assumption is not valid during monsoon. The role of Aitken mode composition and mixing state is very significant in CCN activation during the wet-monsoon conditions.

Even though the aerosol-CCN conditions correspond to a 'polluted-marine' conditions over the rain-shadow region, the very low aerosol loading (towards an aerosol limited regime) during the ISM rainfall, adds the significance of CCN in cloud droplet concentrations. The regional CCN concentration can be estimated from the aerosol size distribution indicating the size dependency of CCN activation during continental (dry) airmass conditions. But the distinct aerosol and CCN properties during the monsoon due to the change in the aerosol source and sink mechanisms lead to the stronger dependence of CCN activation on the composition of Aitken mode aerosols and its mixing state. Aerosols similar to the composition of sulfates





existing in accumulation mode enhances the CCN activation during continental air mass, while the accumulation mode aerosols

during the monsoon have low hygroscopicity. The predominance of ultrafine particles in the boundary layer and the corresponding very low CCN efficiency after the onset of ISM demand further studies using the simultaneous cloud base observations to understand the ACI affecting the precipitation pattern over the rain shadow region against the backdrop of cold phase invigoration (Rosenfeld et al., 2008; Gayatri et al., 2017) and condensational heating (Khain et al., 2012; Fan et al., 2018) mechanisms of tropical convective clouds.

**Appendix**

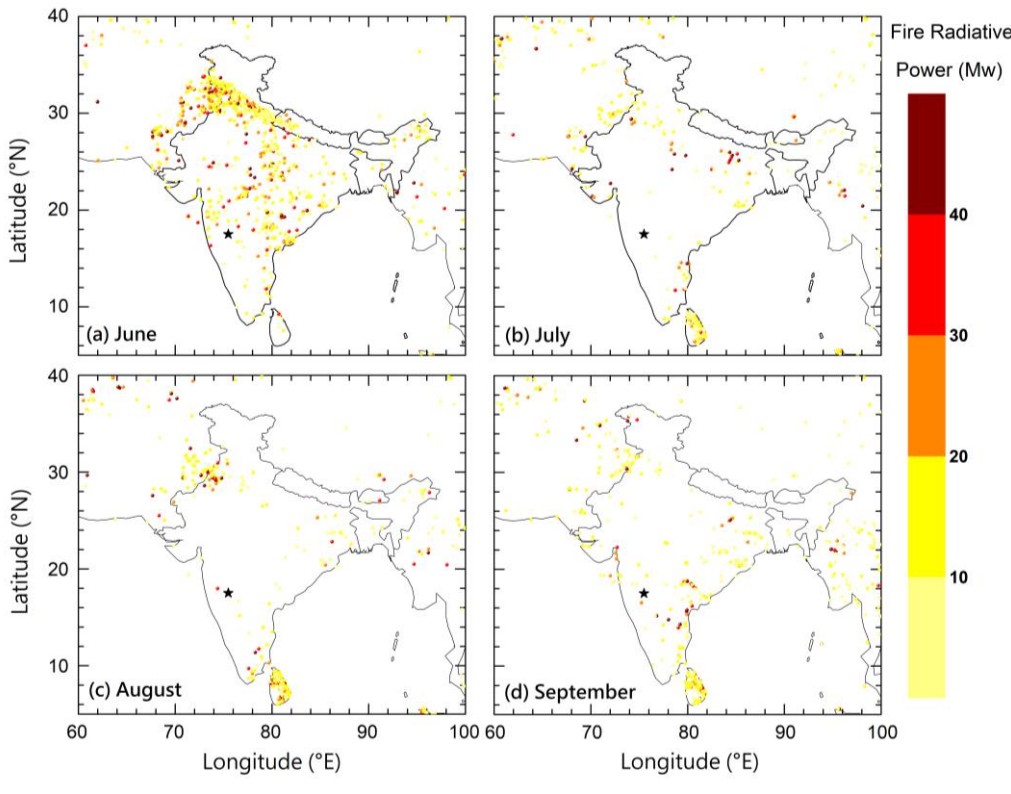

**Figure A1**. Spatial distribution of the Moderate Resolution Imaging Spectroradiometer (MODIS) fire radiative power (Collection 6 product obtained from https://earthdata.nasa.gov/firms) for the measurement periods, along with the observation site marked as a black star.

**Data availability**

Data used in the present study can be obtained by making a request through http://www.tropmet.res.in/~caipeex/ registration form.php or contacting thara@tropmet.res.in.



**Competing interests**

The authors declare that they have no conflict of interest.

**Author contributions**

TVP conceptualized the experiment. TVP and VJ designed the study. TVP, PM, KST, SPB, GD, NM, JR, MK, SD, MV and PDS were responsible for conducting the campaign and data collection. VJ carried out the scientific analysis of the data and drafted the manuscript. TVP carried out the review and editing of the manuscript.

**Acknowledgments**

The CAIPEEX project is funded by the Ministry of Earth Sciences (MoES), Government of India. The authors acknowledge
with gratitude the team effort and dedication of all CAIPEEX team members at the ground observatory. The authors are thankful to the Director and Principal of N. B. Navale Sinhgad College of Engineering for the support. We acknowledge NOAA ARL for the providing the Hybrid Single-Particle Lagrangian Integrated Trajectory (HYSPLIT) transport and dispersion model used in this study. We are thankful to the MODIS team for making AOD data freely available. We acknowledge the use of data from LANCE FIRMS operated by NASA's Earth Science Data and Information System (ESDIS)
with funding provided by NASA Headquarters (http://earthdata.nasa.gov/firms).

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

**Table 1:** Details of aerosol measurements used in the current study

| Sl. No. | Measurements | Instrument | Period (2018) | Reference |
|---|---|---|---|---|
| 1 | CCN concentrations at different SS | CCN counter (CCN-100, make: DMT) | June 1-8 | Roberts and Nenes, (2005) |
| 2 | Aerosol NSD from ~15-655 nm | SMPS (LDMA + CPC, make: TSI) | July 8-12, 15-31 August 1-27, 28-31 | Wiedensohler, (1988) |
| 3 | Aerosol absorption properties at 7 wavelengths | AE33 (make: Magee Scientific) | September 15-30 | Drinovec et al., (2015) |





**Table 2:** CCN and aerosol characteristics reported over various locations along with the present results (at 0.3 % SS). CCN reported for (@ 0.4%, # 0.36 SS

| Location (coordinates; a.m.s.l) | Type/Condition | Period | CN (cm⁻³) | CCN (cm⁻³) | k | AF (%) | GMD (nm) | Reference |
|---|---|---|---|---|---|---|---|---|
| Solapur (17.65° N, 65.9° E, ~480 m) | Continental | Jun 2018 | 3427±1064 | 1946±594 | 0.52±0.11 | 0.55±0.09 | 85 ± 10 | Present Study |
| | Monsoon | Jul 2018 | - | 357±92 | 0.58±0.16 | - | - | |
| | Monsoon | Aug 2018 | 2356±984 | 322±118 | 0.67±0.18 | 0.15±0.06 | 69 ± 11 | |
| | Continental | Sep 2018 | 4381±1824 | 1497±524 | 0.56±0.16 | 0.32±0.10 | 81 ± 14 | |
| Mahabubnagar 17°N, 78°E | Continental (Polluted) | Oct 2011 | - | ~5400 at 1 % SS | ~0.45 | ~0.9 | - | Varghese et al., (2016) |
| Ponmudi (8.8°N, 77.1°E; ~960 m) | Western Ghats Monsoon | Jul-Sep, 2016 | ~2000 | ~400 | 0.65±0.28 | ~0.20 | - | Jayachandran et al., (2018) |
| Mahabaleshwar (17.56°N, 73.4°E; 1348 m) | Western Ghats/ Pre-monsoon Monsoon | Mar-May Jun-Aug, 2012 | ~3100 ~3200 | ~1200 ~500 at 0.2 % SS | ~0.5 ~1 | ~0.35 ~0.35 | ~90 ~77 | Leena et al., (2016) |
| Thumba@ (8.5°N, 76.9°E; 3 m) | Coastal Monsoon | Aug-Sep, 2013 | ~4900 | 2096 ± 834 | 0.54 ± 0.21 | 0.46 ± 0.15 | ~103 | Jayachandran et al., (2017) |
| Nainital (29.2°N, 79.3°E; 1960 m) | Central Himalayas | Jun 2011 | 2425±1112 | 925±601 | 0.57±0.11 | 0.38±0.11 | - | Dumka et al., (2015) |
| | | July 2011 | 1874±776 | 881±500 | 0.45±0.08 | 0.47±0.11 | | |
| | | Aug 2011 | 1606±453 | 684±396 | 0.45±0.04 | 0.42±0.18 | | |
| | | Sep 2011 | 2304±904 | 1233±677 | 0.39±0.03 | 0.54±0.12 | | |
| Darjeeling (27.02°N, 88.25°E; 2200 m) | Eastern Himalayas | Mar-May, 2016 | 7220 ± 1988 | ~1600 | 0.38 ± 0.05 | ~0.25 | - | Roy et al., (2017) |
| Kanpur (26.5°N, 80.3°E; 142 m) | Urban/Polluted | May-Jun (dry) Aug (wet) | ~7110 ~6450 | ~4570 ~2360 | - | ~0.64 ~0.36 | - | Bhattu and Tripathy, 2014 |





| | | | | | | | | |
|---|---|---|---|---|---|---|---|---|
| Korea (37.6 °N, 127.04° E) | Urban/Polluted | May-Jun, 2016 | 10825 ± 4863 | 3105 ± 1521 | - | - | 44 ± 14 | Kim et al., (2018) |
| Guangzhou, (23.07°N, 113.21°E) | Clean Polluted | Summer | 8246±3595 7193±3775 | 3017±1450 2883±1158 | - | 0.39±0.12 0.45±0.13 | - | Duan et al., (2017) |
| Colarado[#] (38.64° N, 105.11° W, 2300 m) | Forest | Jun Jul Aug | ~1400 ~1800 ~1250 | ~500 | - | ~0.30 | ~68 ~80 ~90 | Levin et al., (2012) |
| Amazon 2.13° S, 59° W; 130 m | Forest Dry Wet | Aug-Nov Feb-May | 1520±780 330±130 | ~1469 ~289 at 1% SS | 0.36±0.06 0.57±0.03 | - | - | Pohlker et al., (2016; 2018) |





**Table 3:** The linear fit parameters between BC mass and Aitken/accumulation mode particle concentration

| Period | BC – Aitken particles | | | BC – Accumulation particles | | |
|---|---|---|---|---|---|---|
| | m | C | $R^2$ | m | C | $R^2$ |
| June | 0.43 ± 0.13 | 1224 ± 241 | 0.09 | 2.48 ± 0.14 | -804 ± 165 | 0.74 |
| August | 0.41 ± 0.03 | 503 ± 35 | 0.28 | 1.32 ± 0.07 | 133 ± 48 | 0.38 |
| September | 0.63 ± 0.05 | 446 ± 114 | 0.37 | 1.02 ± 0.06 | -9 ± 111 | 0.53 |