# Peer review of "CCN characteristics during the Indian Summer Monsoon over a rainshadow region"

_Atmospheric Chemistry and Physics, 2020_

## Referee Comment (RC1) · Anonymous Referee #1 · 17 Mar 2020

The manuscript "CCN characteristics during the Indian Summer Monsoon (ISM) over a rainshadow region" by Jayachandran et al. presents a comprehensive study of CCN characteristics of aerosol particles in the Indian sub-continent prior to, during and after the Indian Summer Monsoon. The paper discusses on-line particle measurements, including CCN number concentrations and aerosol particle size distributions, as well as the CCN parameters derived from these measurements. The aerosol data are supplemented by the aethalometer and meteorological data, as well as HYSPLIT trajectories.

At the moment, the presented study is of rather limited scientific relevance as the potential for CCNC in aerosol-cloud interactions studies has pretty much been exhausted. This is exemplified by the fact that the majority of referenced literature is at least a decade old or more. At this point, CCNC can no longer help us understand aerosol-cloud interactions, and the majority of presented outcomes are already known. This notion is not meant to change or take away from the paper; it is more meant as the direction of potential future work for the authors.

Having said that, the paper is of very good quality and written very well, and the authors do an excellent job at interpreting the data, presenting the results and discussing them in detail (sometimes too much detail). The objectives and conclusions are clearly stated, and the paper makes great use of existing literature and puts its results in perspective. The paper is well-structured and provides the interested reader with a lot of information about CCN characteristics in India during ISM. The authors make as much use of the data as I think is possible, which is definitely a benefit of the presented study. At some points throughout the paper authors make claims that are not supported by their observations, and the paper overall is quite long. However, I definitely recommend the manuscript to be published after the minor revisions suggested below are incorporated.

**General comments**
1. Lines 40-41 and 594-595 – We know now that both of these statements are not true. CCN characteristics of aerosol particles (size, chemistry, etc.) have now been described in many locations all over the world, yet challenges in understanding aerosol-cloud interactions (ACI) remain. The biggest challenges in ACI are understanding how CCN interact with water vapour in real atmosphere. We know very little about actual ambient supersaturation levels and the depletion of water vapour during CCN activation, something CCNC cannot help us with. Additionally, there is a very large disconnect between ambient CCN and cloud droplet number concentration (CDNC) (Moore et al., 2013), something we also fully don't understand. Please, rephrase or remove the statements.

2. In the overwhelming majority of CCN-related and referenced literature, k-value, or κ-value, is predominantly used to describe the hygroscopicity parameter kappa κ (Petters and Kreidenweis, 2007). In the current manuscript, it connotates an empirical fit value of the Twomey's fit. I found it very confusing while reading through the paper, especially since the magnitudes of both parameters are very similar. Is it possible to use any other connotation for the empirical fit value? I think it would make it clearer what parameter you are referring to, but I leave the decision at the discretion of the authors.

3. There are three main periods discussed in this study – pre-Monsoon, Monsoon and post-Monsoon. Sometimes they are referred to as such. Sometimes they are referred to by the month. Sometimes they are referred to by continental and marine airmass. Sometimes they are referred to as dry and wet conditions. I found it confusing and I always had to go back and

check which period is meant. I think it would make the paper a lot easier to read if the authors stick to one way of describing these periods.

4. The paper should acknowledge more the fact that during July and August a lot of the aerosol particles and potentially good CCN are either washed out by wet scavenging or have already activated into cloud droplets, thereby in both cases being removed from the measured ambient aerosol population. In this sense, the aerosol properties measured in July and August represent a subset of APs that is already inherently CCN-inactive. The paper alludes to this on several occasions, but I think it should be present more throughout the paper. For example, lines 497-498 – how do you know that low GMD and high $d_{cri}$ in August indicate the presence of freshly emitted water-inactive primary organic aerosols? Maybe your larger accumulation mode particles were simply removed by deposition/activation, leaving fine particles behind, skewing the NSD towards lower sizes and increasing your $d_{cri}$.

5. Lines 499-524 – This whole section can be removed from the paper as it presents little to no new or exciting information. Of course, AF and $d_{cri}$ are anticorrelated. It makes perfect sense and wouldn't be any other way. Please, consider removing this section and starting a new section 3.6 at line 525 to describe the relationship between aerosol absorption and CCN properties.

6. Section 3.6 – In this section authors make an attempt to estimate the CCN concentration using several parameters, including a cut-off size, critical diameter or some predefined chemical information. It is immediately visible that setting a cut-off size alone does an excellent job in estimating CCN concentration. All R values are 0.95 or higher, which is amazing and unlikely to get any better. We already know well that size matters way more than chemistry (Dusek et al., 2006), and the fact that you have a >90% predicting capacity of CCN concentration simply by setting a lower size limit is very good and really all we need to know. Especially, since the correlations get even worse when you use $d_{cri}$ or chemical information. The discussion in section 3.6 needs to be reduced by quite much to highlight that particle size is more than enough to accurately estimate CCN concentration. There is absolutely no need to discuss and explain why R of 0.95 is worse than 0.97 because both of these values are very high, higher than in many other studies. The authors seem to be very perfectionist in this section and make statements that are not supported by observations. Chemistry does not play a crucial role in determining CCN efficiency, and there is no need to try and convince the reader that it does. The last bullet point in the Summary and Conclusions section should reflect this as well.

7. Lines 703-714 – the discussion here needs to be changed to account for the facts that a) most of CCN-active particles during Monsoon are already removed by activation/deposition and what's left is inherently CCN-inactive, and b) size alone is sufficient to accurately estimate CCN concentration during all months and conditions of the performed study.

**Minor comments**
1. Line 155 – please, define the observation period. It is seen in Table 1 and mentioned in the abstract, but I think it would be nice to include it in the main text as well as well.

2. Lines 162-164 – the sentence makes no sense. Please, rephrase.

3. Line 180 – winds were blowing from the north-east only during September. Please, state that.

4.  Lines 204-205 – "during (b) prior (June) and (c) after (September) monsoon" part makes no sense. Please, rephrase. Also, "…white star *indicates*…".

5.  Please, be consistent with units of measurement of BC concentration. Either ng or µg.

6.  Lines 231-232 – What is the reason for the difference in CCN concentration between continental and marine airmasses? Is it simply because the total number of all AP is different? Or because CCN are less hygroscopic during ISM?

7.  Line 247 – that is not really true. Jun, Jul and Sep values are basically the same, only Aug value is higher. But their variability (±) is high, and I would say there is no difference. This should be reflected in the discussion elsewhere, e.g. line 266.

8.  Line 281 – there is also an increase in CCN after sunrise in September (Fig. 5b). The increase is just not as dramatic as in other months.

9.  Figure 6 – have you tried combining panels a and b, and panels c and d? It would make comparing them much easier and different magnitudes of values would be easier to see. I have to carefully look at the y-axis values to see that the variation is less/more prominent.

10. Line 288 – I would say that a small increase in CN is seen in the afternoon, instead of around mid-noon.

11. Figures 5 and 6 – so why is there a second peak in CN in the evening in all months, but there is no corresponding peak in CCN in the evening? What are those CCN inactive particles?

12. Lines 312-313 – the statement is not true! For all three parameters (CCN, CN and BC) the diurnal variation during Jun and Sep is higher than in Jul and Aug.

13. Lines 318-391 – neither are true. CCN is just a fraction of CN, so if CN increases, CCN is likely going to increase as well. Also, all three datasets in Fig. 7 are fitted well with a linear fit, with all $R^2$ values above 0.8. Therefore, the dependence of CCN on total AP in your study is very much linear.

14. Line 334 – The CCN/CN relationship is August is not weak; at $R^2$ of 0.82 it's quite strong actually. It's just weaker than in Jun and Sep.

15. Lines 341-342 – neither statements are true. The variation is linear, as mentioned above, and concentration is not 600 cm$^{-3}$. In Fig. 7b most data points are below 500 cm$^{-3}$, and Fig. 5a clearly shows CCN concentrations in August between 200 and 400 cm$^{-3}$.

16. Lines 358-359 – the sentence is redundant. Basic physics tell us that is SS goes up, so will CCN and AF.

17. Lines 409-414 – biomass burning aerosol is not known to be particularly CCN-active, unless properly aged. The discussion here makes it sound as though high CCN AF is associated with biomass burning, which I don't think is true. Please, rephrase.

18. Lines 423-424 – please, indicate which reference you used for defining nucleation, Aitken and accumulation mode sizes.

19. Lines 439 – instead of saying "in the mean picture", please, refer to the Figure and the panel in question.

20. Line 457 – Figure 11a shows that 81% of particles are below 100 nm in diameter in August, not 60% as mentioned in the text.

21. Lines 468-469 – I don't think all accumulation mode particles are always associated with biomass burning. Or maybe better to say that accumulation mode particles are associated *either* with biomass burning *or* with condensation and coagulation of smaller particles.

22. Lines 473-476 – please, use punctuation in this sentence. It is currently not clear if oxygenated organic aerosol and sulfates are responsible for higher GMD or if long chain hydrocarbons are responsible for fine mode and oxygenated organic aerosol.

23. Lines 531-533 – there is no need to describe in the main text what should be and already is in the legend of the figure. All descriptions of the figures (symbols, lines, etc.) should be found in the legend and not in the main text. Please, correct this throughout the entire manuscript. This will also make the paper a bit shorter.

24. Line 534 – "comparatively high" should be replaced with "moderate".

25. Figure 14 – is impossible to read. First, what are the colours and symbols? The legend needs to be improved so it is clear what symbols and colours mean. Second, I would say that the BC mass is the independent variable and Aitken mode particles are the dependent variable, so the axes in the figure need to be switched.

26. Line 676 – when is this true? During ISM?

**Technical comments**
There are several grammatical, punctuation and other errors, most of which will be corrected during the copy-editing stage of the manuscript. The errors indicated below stood out but are not exhaustive.

1. Lines 73-77 – the sentence is missing a verb.

2. Line 93 – "…data presented in this study *are*…". Data are plural and this should be reflected everywhere else in the paper.

3. Line 177 – remove the word "months" after "July and August"

4. Line 187 – "CCN characteristics at the site *are*…"

5. Line 266 – "An enhancement in k-values *is*…"

6. Line 356 – should say "*inactive*"

7. Line 436 – "…*are* also similar…"

8. Line 565 – my name is misspelled. Should be Paramonov ☺

**References**

- Dusek, U., Frank, G. P., Hildebrandt, L., Curtius, J., Schneider, J.,Walter, S., Chand, D., Drewnick, F., Hings, S., Jung, D., Borrmann, S., and Andreae, M. O.: Size matters more than chemistry for cloud-nucleating ability of aerosol particles, Science,312, 1375–1378, doi:10.1126/science.1125261, 2006.
- Moore, R. H., Karydis, V. A., Capps, S. L., Lathem, T. L., and Nenes, A.: Droplet number uncertainties associated with CCN: an assessment using observations and a global model adjoint, Atmos. Chem. Phys., 13, 4235–4251, doi:10.5194/acp-13-4235-2013, 2013.
- Petters, M. D. and Kreidenweis, S. M.: A single parameter representation of hygroscopic growth and cloud condensation nucleus activity, Atmos. Chem. Phys., 7, 1961–1971, doi:10.5194/acp-7-1961-2007, 2007.

Thank you for an excellent paper and best of luck with the review process!

---

## Referee Comment (RC2) · Anonymous Referee #2 · 18 Mar 2020

Review comments on the manuscript "CCN characteristics during the Indian Summer Monsoon over a rain shadow region" by Jayachandran et al., 2020

General: The article provides a comprehensive account of cloud condensation nuclei characteristics over a rain shadow region in Western Ghats India. CCN study over Indian region, especially in the rain shadow regions are important in understanding aerosol-cloud interactions and their implications. The data collection and analysis are quite extensive and results are presented comprehensively. I recommend publishing this work after the comments are adequately addressed. One general lacuna is that the authors do not go beyond reporting the data and results of analysis, which though

are good in themselves. A rigorous discussion in the light of the results on the CCN characteristics is needed to improve the scientific content in this work. In general, an improvement of the language would help to understand the importance of the finding better.

Major Comments:

1. Entire analysis of this study is a comparison between CCN characteristics during continental air mass and marine air mass during 2018 monsoon over a rain shadow region in western ghat, India. However, it is not clear that how they delineated the continental air mass and marine air mass trajectories. It is important to make to clear whether the Hysplit model was ran for every 30 min (since the CCN and other data are available for 30 min interval) and the data are segregated accordingly for analysis for the entire study period or took a specific time and used that data only for further analysis. For a general reader it seems that entire June and September trajectories over study region are continental. But it is also mentioned that the monsoon onset is on 08 June.

2. Most of the analysis focuses on reporting of values and comparison with reported values from other sites. Authors should also try to add more science through discussion related to the implication of their observation. For instance, discussion on the implication of the role of carbonaceous aerosols in acting as CCN over the study region during dry conditions leading to semi-direct effect/rapid adjustment. Does dust aerosol have any role in modulating CCN properties over the region?

3. What hypothesis the authors put forward to explain the reduced activation ratio during monsoon/ marine airmass conditions, when normally the aerosols would be richer in hygroscopic species, that could be easily activated?

4. In CCN closure analysis, describe the methodology and assumptions used in estimating CCN. Why aerosol composition is assumed to be ammonium sulphate? In the analysis authors have explicitly tried to establish the influence of carbonaceous

aerosols in acting as a CCN?

Minor Comments Line 32-40: Sentence is confusing and needs modification. It gives a feel that "Condensation nuclei" and "Cloud Condensation nuclei" are same. The sentence starting with "For a fixed liquid water content. . . . . ." need to be revised.

Line 40-45: "Characterization of CCN . . . . . . . . . . . . . . the physical and chemical characteristics of AP". These two sentences can be reframed to one as it tries to convey the same information.

Line 52-55: "For a given particle . . . . . . . . . . . .the accuracy of climate models to address the ACI (Fountakis and Nenes,2005)". The relevance of this sentence in the paragraph is not understood. In the second paragraph authors try to portray the heterogeneities of aerosol particles and CCN in the global scenario as well as in Indian context. Authors can discuss more on the role of organics as CCN as they can reveal the first indirect effect (Nenes et al., 2002) and as well as studies over organics in Indian context.

Line 73-77: "Various studies. . . . . . . . . .. from the unique data obtained from the CAIPEEX". Rephrase the sentence.

Line 80-90: Authors have mentioned that a few studies (Leena et al., 2016, Jayachandran et al., 2018) have already reported CCN characteristics over different locations of Western Ghats. If so, does this study address the same objectives with observations form a different site? Please bring more clarity to the objectives of this study.

Line 110: Were the data corrected for the maximal activated fraction, which is of high importance, in particular for total CCN measurements (Paramonov et al., 2013; Rose et al., 2010)? Please give more information of reference data used in the köhler theory when performing the CCN calibration. This is very important because different parameterizations will retrieve different critical supersaturations (Rose et al.,2008; Wang et al., 2017). Also mention the uncertainty in measurements of different instruments.

Line 162-165: Rewrite the captions specifically for Figure1. (a) & (b). Line 186: Correct

bullet numbering.

Line 190-220: There are several concerns in this analysis: (1) Does it mean that for all days in June and September, trajectories ending over study region were of continental origin? This is difficult to comprehend especially when authors have mentioned that monsoon onset over study region was on 08 June2018. (2) AOD retrieved from MODIS over land and especially during ISM is a matter of concern. (3) Authors also mentioned about MODIS retrieved fire count information, please do mention the confidence level used as well as its uncertainty.

Line 225-270: From this study as well as those conducted over Mahabaleshwar and Amazon, an increase in ïĄń values is reported during wet months. What is the scientific reason?

Line 315-350: Justify your arguments on the formation of NPF over the study region during wet conditions? Why the CN-CCN relationship weakens during September? CN-CCN relationship seems to hold strong when CN concentration is ∼3.7*10ˆ3 particles/cm-3. Is it due to instrumental artefact? Or do you propose any process?

Line 360: Figure 8 clearly shows that activation fraction (SS) is very low during wet months. What does it imply? Are similar observations are reported elsewhere? Discuss more on the implication and physical mechanisms?

Line 365-370: Why the diurnal variation in Twomey's empirical fit parameter –ïĄń and activation fraction is not showing (revere) relationship that is obviously seen in other months. During September, diurnal variation in GMD showed morning high values, which is also reflected in AF but not in Twomey's empirical fit parameter –ïĄń. Similarly, in June, GMD showed low values during morning hours without any significant change in AF and Twomey's empirical fit parameter –ïĄń. Discuss the scientific implications of these

Line 395-400: Authors have tried to associate high Twomey's empirical fit parameter

–ïĄń values observed during morning hours of wet months to organic aerosol mostly produced by biomass burning. However, the MODIS fire count map during August shows very less fire count over the study region. Justify the statement?

Line 400-414: "Thus, the aerosol composition especially the organic aerosols….." Please justify. In literature AAE greater than 2 is usually inferred as biomass source and AAE between 1 and 2 is usually considered as mixture of BC and OC (Bergstorm et al., 2007). Authors should present independent observations or data to attribute this to organic aerosols.

Line 515: Are the estimated 'a' and 'b' value are site and season specific. ?

Line 555: What about the role of dust aerosol (local/transported) acting as CCN? As can be seen in Figure, there are a few points where the AAE is less than 1. Some studies have attributed such points to dust coated with BC. Do back-trajectories in these case support dust transport?

Line 590: Figure 13 and Figure 14 clearly indicates the role of carbonaceous aerosols acting as CCN over the study region during dry conditions (June and September). Is there any specific reason for the better correlation observed in June?

Straighten up the formatting errors in reference list. Check line 855 for example.
* * *

---

## Author Comment (AC1) · 12 May 2020

We thank the Editor and appreciate the evaluation of our results by both the referees. We think that the recommendations lead to the overall improvement of the manuscript. We have carefully considered the comments and suggestions, and revised the paper accordingly. Our point-by-point responses to the comments, based on which the revisions are made, are given below. The review comments are given in italics, while the author responses are in bold font

**Anonymous Referee #1**

*The manuscript "CCN characteristics during the Indian Summer Monsoon (ISM) over a rainshadow region" by Jayachandran et al. presents a comprehensive study of CCN characteristics of aerosol particles in the Indian sub-continent prior to, during and after the Indian Summer Monsoon. The paper discusses on-line particle measurements, including CCN number concentrations and aerosol particle size distributions, as well as the CCN parameters derived from these measurements. The aerosol data are supplemented by the aethalometer and meteorological data, as well as HYSPLIT trajectories.*

*At the moment, the presented study is of rather limited scientific relevance as the potential for CCNC in aerosol-cloud interactions studies has pretty much been exhausted. This is exemplified by the fact that the majority of referenced literature is at least a decade old or more. At this point, CCNC can no longer help us understand aerosol-cloud interactions, and the majority of presented outcomes are already known. This notion is not meant to change or take away from the paper; it is more meant as the direction of potential future work for the authors.*

*Having said that, the paper is of very good quality and written very well, and the authors do an excellent job at interpreting the data, presenting the results and discussing them in detail (sometimes too much detail). The objectives and conclusions are clearly stated, and the paper makes great use of existing literature and puts its results in perspective. The paper is well-structured and provides the interested reader with a lot of information about CCN characteristics in India during ISM. The authors make as much use of the data as I think is possible, which is definitely a benefit of the presented study. At some points throughout the paper authors make claims that are not supported by their observations, and the paper overall is quite long. However, I definitely recommend the manuscript to be published after the minor revisions suggested below are incorporated.*

**We thank the reviewer for the encouraging comments and fruitful suggestions.**

**General comments**

*1. Lines 40-41 and 594-595 – We know now that both of these statements are not true. CCN characteristics of aerosol particles (size, chemistry, etc.) have now been described in many locations*

*all over the world, yet challenges in understanding aerosol-cloud interactions (ACI) remain. The biggest challenges in ACI are understanding how CCN interact with water vapour in real atmosphere. We know very little about actual ambient supersaturation levels and the depletion of water vapour during CCN activation, something CCNC cannot help us with. Additionally, there is a very large disconnect between ambient CCN and cloud droplet number concentration (CDNC) (Moore et al., 2013), something we also fully don't understand. Please, rephrase or remove the statements.*

**Agreeing with the reviewer. Lines 40-41 is modified as follow, 'Characterization of the hygroscopic growth of AP, which is generally addressed by the Köhler theory (Köhler, 1936), is the most fundamental aspect in assessing the aerosol-cloud interactions (ACI) for reducing the uncertainties in indirect radiative forcing estimation'.**

**'In the real atmosphere, the supersaturation measurements are seldom possible and the large disagreements between the CCN and cloud droplet number concentration remains elusive (Moore et al., 2013).'**

**and lines 594-595 are removed.**

*2. In the overwhelming majority of CCN-related and referenced literature, k-value, or κ-value, is predominantly used to describe the hygroscopicity parameter kappa κ (Petters and Kreidenweis, 2007). In the current manuscript, it connotates an empirical fit value of the Twomey's fit. I found it very confusing while reading through the paper, especially since the magnitudes of both parameters are very similar. Is it possible to use any other connotation for the empirical fit value? I think it would make it clearer what parameter you are referring to, but I leave the decision at the discretion of the authors.*

**We understand the concern. Since all the papers using the Twomey's empirical fit have used the 'k' notation we are also following the same. Considering the reviewer's suggestion, we are clarifying this usage in the manuscript at Line 237 as,**

**'It should be noted that, the empirical fit parameter k is different from the effective hygroscopicity parameter-κ discussed by Petters and Kreidenweis (2007).'**

*3. There are three main periods discussed in this study – pre-Monsoon, Monsoon and post- Monsoon. Sometimes they are referred to as such. Sometimes they are referred to by the month. Sometimes they are referred to by continental and marine airmass. Sometimes they are referred to as dry and wet conditions. I found it confusing and I always had to go back and check which period is meant. I think it would make the paper a lot easier to read if the authors stick to one way of describing these periods.*

**All the analysis in the current manuscript are based on the continental and marine airmass trajectories which coincides with the dry and wet conditions over the region, respectively, within the Indian Summer Monsoon period (June to September) of 2018. Though months were mentioned, it represents the corresponding days only which**

experienced the marine/continental air mass. However, to avoid the confusion the periods are uniformly referred as continental (1 and 2) and marine (1 and 2) conditions in the modified manuscript. Continental-1 and 2 comprise of 1-8 June and 15-30 September. While, the marine-1 and 2 comprises of 8-12, 15-31 of July and 1-27. 28-31 of August. These changes are implemented in the whole manuscript including figures.

**These clarifications are mentioned in Lines 155-161**

*4. The paper should acknowledge more the fact that during July and August a lot of the aerosol particles and potentially good CCN are either washed out by wet scavenging or have already activated into cloud droplets, thereby in both cases being removed from the measured ambient aerosol population. In this sense, the aerosol properties measured in July and August represent a subset of APs that is already inherently CCN-inactive. The paper alludes to this on several occasions, but I think it should be present more throughout the paper. For example, lines 497- 498 – how do you know that low GMD and high dcri in August indicate the presence of freshly emitted water-inactive primary organic aerosols? Maybe your larger accumulation mode particles were simply removed by deposition/activation, leaving fine particles behind, skewing the NSD towards lower sizes and increasing your dcri.*

**We agree with the reviewer.**

**The missing of CCN active aerosol particles by wet scavenging or activation to cloud droplets leading to the low CCN activation fraction is now highlighted in the manuscript at line 705 as,**

**'In the presence of marine airmass trajectory (July and August), most of the fine AP which are potential CCN are either washed out by wet scavenging or already activated as cloud droplets. There is also possibility of less emissions due to wet conditions. Hence the measured AP are devoid of CCN active particles which are clearly seen from the aerosol NSD during the relevant periods. Thus, the low CCN activation fraction during the marine conditions is due to the missing of those CCN active particles near the surface.'**

**The drastic daytime increase in the absorption Angstrom exponent especially during marine conditions as seen in Fig 10(a) is considered as an indication for the local sources contributing to the primary organic aerosols. However, in the current paper those aspects cannot be proved and will be investigated in the ongoing work from the location. Hence, Lines 497-498 are removed.**

*5. Lines 499-524 – This whole section can be removed from the paper as it presents little to no new or exciting information. Of course, AF and dcri are anticorrelated. It makes perfect sense and wouldn't be any other way. Please, consider removing this section and starting a new section 3.6 at line 525 to describe the relationship between aerosol absorption and CCN properties.*

**As per suggestion, this part of the manuscript is removed in the revision.**

*6. Section 3.6 – In this section authors make an attempt to estimate the CCN concentration using several parameters, including a cut-off size, critical diameter or some predefined chemical information. It is immediately visible that setting a cut-off size alone does an excellent job in estimating CCN concentration. All R values are 0.95 or higher, which is amazing and unlikely to get any better. We already know well that size matters way more than chemistry (Dusek et al., 2006), and the fact that you have a >90% predicting capacity of CCN concentration simply by setting a lower size limit is very good and really all we need to know. Especially, since the correlations get even worse when you use dcri or chemical information. The discussion in section 3.6 needs to be reduced by quite much to highlight that particle size is more than enough to accurately estimate CCN concentration. There is absolutely no need to discuss and explain why R of 0.95 is worse than 0.97 because both of these values are very high, higher than in many other studies. The authors seem to be very perfectionist in this section and make statements that are not supported by observations. Chemistry does not play a crucial role in determining CCN efficiency, and there is no need to try and convince the reader that it does. The last bullet point in the Summary and Conclusions section should reflect this as well.*

**Thanks for the suggestions.**

**As the reviewer rightly pointed, the estimated CCN showed good correlation in all the three methods, indicating the prominence of aerosol size in CCN activation. Still, there is a significance difference in the activation efficiency of accumulation mode AP prior to and post the monsoon, indicating the difference in the composition. Similarly, the over-estimation of CCN concentration during marine condition for accumulation mode (2) and ammonium sulfate composition assumptions (3.38) shows the hygrophobic composition of those AP. The discussions in section 3.6 is modified as per the reviewer's suggestion and the conclusions are also modified.**

7. Lines 703-714 – the discussion here needs to be changed to account for the facts that a) most of CCN-active particles during Monsoon are already removed by activation/deposition and what's left is inherently CCN-inactive, and b) size alone is sufficient to accurately estimate CCN concentration during all months and conditions of the performed study.

**Complied with. The lines are modified as follows,**

**'The closure study indicates the size dependency of CCN activation especially during dry-continental conditions. Most of the CCN-active (fine) AP were removed from atmosphere by activation or wet removal and the remained particles were inherently CCN-inactive as seen in the aerosol NSD during the marine air mass.'**

**Minor comments**

1. *Line 155 – please, define the observation period. It is seen in Table 1 and mentioned in the abstract, but I think it would be nice to include it in the main text as well as well.*

**Complied with.**

2. *Lines 162-164 – the sentence makes no sense. Please, rephrase.*

**Lines 162-164 modified as,**

**'From Figure 1(a), it can also be seen that the air mass history for the continental classification are mostly within 2 km above the surface, indicating the chances for the influence of local aerosol sources.'**

3. *Line 180 – winds were blowing from the north-east only during September. Please, state that.*

**Figure 2(f) shows that continental air mass days in September experienced winds from both North-East and South-West. However, Line 180 is modified as,**

**'Westerly and South-Westerly winds were present during marine 1 and 2 conditions (Fig. 2d and e), while the continental-2 days (Fig. 2f) mostly North-Easterly winds were observed.'**

4. *Lines 204-205 – "during (b) prior (June) and (c) after (September) monsoon" part makes no sense. Please, rephrase. Also, "…white star indicates…".*

**The lines are modified as,**

**'Aerosol optical depth (AOD) at 550 nm observed from the Moderate Resolution Imaging Spectroradiometer (MODIS) for the continental (a) 1 and (b) 2 conditions. The site-Solapur is indicated by the white star in the spatial AOD maps.'**

5. *Please, be consistent with units of measurement of BC concentration. Either ng or µg.*

**Suggested changes are implanted in the revision. All the units are now represented in ng m$^{-3}$.**

6. *Lines 231-232 – What is the reason for the difference in CCN concentration between continental and marine airmasses? Is it simply because the total number of all AP is different? Or because CCN are less hygroscopic during ISM?*

**The study investigates this difference in the CCN population within the ISM period which coincide with different air mass history conditions. The analyses are meant to unravel the causes and found that, the aerosols present during the wet conditions are those CCN-inactive particles mainly due to the wet scavenging. The accumulation mode AP during the marine air mass was less hygroscopic. The distinct aerosol NSD during the marine and continental air mass conditions underline these findings.**

*7. Line 247 – that is not really true. Jun, Jul and Sep values are basically the same, only Aug value is higher. But their variability (±) is high, and I would say there is no difference. This should be reflected in the discussion elsewhere, e.g. line 266.*

**We agree with the reviewer. Line 247 is removed and the corresponding discussions at Line 266 is modified as,**

**'Generally, an enhancement in k-values is observed during the monsoon period, which is seen only during the marine-2 conditions in this study. In all other cases, the k-values are comparable.'**

*8. Line 281 – there is also an increase in CCN after sunrise in September (Fig. 5b). The increase is just not as dramatic as in other months.*

**Thanks for the suggestion. This point is added in the manuscript at Line 281.**

9. Figure 6 – have you tried combining panels a and b, and panels c and d? It would make comparing them much easier and different magnitudes of values would be easier to see. I have to carefully look at the y-axis values to see that the variation is less/more prominent.

**Suggested changes are implanted in the revision.  Figure 6 is modified as the following,**

[Figure]

*10. Line 288 – I would say that a small increase in CN is seen in the afternoon, instead of around mid-noon.*

**Suggested changes are implemented in the revision.  'Around mid-noon' is replaced with 'afternoon' in the manuscript**

*11. Figures 5 and 6 – so why is there a second peak in CN in the evening in all months, but there is no corresponding peak in CCN in the evening? What are those CCN inactive particles?*

**As the reviewer rightly pointed out, the evening peak observed in the CN concentration is not prominent in CCN diurnal variations, except during the marine air mass of July. Interestingly the CCN concentration peaked with sunrise and remained high during the daytime, especially during the marine airmass conditions. To investigate this pattern, the diurnal variation of nucleation mode particles is examined and is shown below, since the size mostly governs the CCN activation. From the figure, it is clear that the evening peak in CN concentration (August and September) is associated with an enhancement in the nucleation mode AP, which do not contribute to the CCN concentration.**

[Figure]

*12. Lines 312-313 – the statement is not true! For all three parameters (CCN, CN and BC) the diurnal variation during Jun and Sep is higher than in Jul and Aug.*

**We have examined the normalised diurnal variations (normalised with the least value in a day) and the coefficient of variance (CV) of CCN, CN and BC concentrations, and shown below. BC showed a clear diurnal variation under all the conditions. All the three parameters showed diurnal variations during the continental air mass (2) in September. Other than those, during the marine air mass (2) in August, both CCN and BC had a clear diurnal variation. But the CN concentration shows a prominent diurnal variation for the continental-1 also. Though the diurnal variations in BC was high during both continental conditions, the corresponding CV was also high, indicating the larger spread of the mean values. Thus, it is difficult to attribute the diurnal variations to marine or continental conditions. Hence, we are modifying the lines 312-313 as,**

**'The diurnal variations during the continental conditions indicate the consistently high AP background conditions. While the diurnal variations during the marine conditions indicate the significant presence of local AP sources.'**

[Figure]

*13. Lines 318-391 – neither are true. CCN is just a fraction of CN, so if CN increases, CCN is likely going to increase as well. Also, all three datasets in Fig. 7 are fitted well with a linear fit, with all R2 values above 0.8. Therefore, the dependence of CCN on total AP in your study is very much linear.*

**Suggested changes are implanted in the revision. The sentence is removed.**

*14. Line 334 – The CCN/CN relationship is August is not weak; at R2 of 0.82 it's quite strong actually. It's just weaker than in Jun and Sep.*

**Suggested changes are implanted in the revision. The sentence is modified as,**

**'The relationship between CCN and CN during marine-2 is weaker than the continental phases, and only a few AP are activating as CCN.'**

*15. Lines 341-342 – neither statements are true. The variation is linear, as mentioned above, and concentration is not 600 cm-3. In Fig. 7b most data points are below 500 cm-3, and Fig. 5a clearly shows CCN concentrations in August between 200 and 400 cm-3.*

**Agree. The variation is linear. The line is modified as,**

**'the CCN concentration at 0.3 % was not increasing beyond 600 cm$^{-3}$, despite of CN concentration increasing to ~7500 cm$^{-3}$.'**

*16. Lines 358-359 – the sentence is redundant. Basic physics tell us that is SS goes up, so will CCN and AF.*

**Suggested changes are implanted in the revision.  The sentence is removed.**

*17. Lines 409-414 – biomass burning aerosol is not known to be particularly CCN-active, unless properly aged. The discussion here makes it sound as though high CCN AF is associated with biomass burning, which I don't think is true. Please, rephrase.*

**Suggested changes are implemented in the revision.  Line 409-414 is modified as,**

**'The high CCN AF during the continental conditions at Solapur is similar to those reported during dry conditions in Nainital (Gogoi et al., 2015), where the high CCN AF was attributed to biomass burning.'**

*18. Lines 423-424 – please, indicate which reference you used for defining nucleation, Aitken and accumulation mode sizes.*

**Suggested changes are implanted in the revision.  We followed the classifications from Ueda et al., (2016) and Willis et al., (2016). The references are added in the manuscript.**

*19. Lines 439 – instead of saying "in the mean picture", please, refer to the Figure and the panel in question.*

**Suggested changes are implanted in the revision.  Line 439 is corrected as,**

**'Three modes are distinctly observed in Figure 11(c), with two peaks below 100 nm'**

*20. Line 457 – Figure 11a shows that 81% of particles are below 100 nm in diameter in August, not 60% as mentioned in the text.*

**For August, the fine particles (<100 nm) are 69%. It is corrected in the manuscript**.

*21. Lines 468-469 – I don't think all accumulation mode particles are always associated with biomass burning. Or maybe better to say that accumulation mode particles are associated either with biomass burning or with condensation and coagulation of smaller particles.*

**Suggested changes are implanted in the revision.  The sentence is modified as,**

**The accumulation mode particles are associated with either aged biomass burning particles (Kalvitis et al., 2015) or condensation and coagulation of smaller secondary organics and inorganics particles (Seinfeld and Pandis, 2016)**

*22. Lines 473-476 – please, use punctuation in this sentence. It is currently not clear if oxygenated organic aerosol and sulfates are responsible for higher GMD or if long chain hydrocarbons are responsible for fine mode and oxygenated organic aerosol.*

**Suggested changes are implemented in the revision.  The sentence corrected as,**

**'while oxygenated organic aerosols and sulfates are responsible for the higher GMD'**

*23. Lines 531-533 – there is no need to describe in the main text what should be and already is in the legend of the figure. All descriptions of the figures (symbols, lines, etc.) should be found in the legend and not in the main text. Please, correct this throughout the entire manuscript. This will also make the paper a bit shorter.*

**Suggested changes are implanted in the revision.**

*24. Line 534 – "comparatively high" should be replaced with "moderate".*

**Suggested changes are implanted in the revision.**

*25. Figure 14 – is impossible to read. First, what are the colours and symbols? The legend needs to be improved so it is clear what symbols and colours mean. Second, I would say that the BC mass is the independent variable and Aitken mode particles are the dependent variable, so the axes in the figure need to be switched.*

**Suggested changes are implemented in the revision. The figure is modified as below and the linear fits are carried out separately for BC concentrations 2500 ng m⁻³, and above it which is considered as the polluted case. This classification is made since the BC mass during the wet periods was mostly (99%) below 2500 ng m⁻³.**

[Figure]

*26. Line 676 – when is this true? During ISM?*

**The lowest values of CCN concentration is reported during the marine air mass conditions during the ISM period over Solapur.**

**Technical comments**

*There are several grammatical, punctuation and other errors, most of which will be corrected during the copy-editing stage of the manuscript. The errors indicated below stood out but are not exhaustive.*

1. *Lines 73-77 – the sentence is missing a verb.*

**Suggested changes are implanted in the revision.  The following revision is done,**

**'Various studies have addressed the spatio-temporal distribution of AP (Padmakumari et al., 2013; Varghese et al., 2019), cloud microphysics (Prabha et al., 2011; 2012; Padmakumari et al., 2018), rainfall (Maheshkumar et al., 2014) properties, the relationship between cloud microphysics and thermodynamics (Bera et al., 2019), and ACI (Pandithurai et al., 2012; Prabha et al., 2012; Konwar et al., 2012; Gayatri et al., 2017; Patade et al., 2019) from the unique data obtained from the CAIPEEX.'**

2. *Line 93 – "…data presented in this study are…". Data are plural and this should be reflected everywhere else in the paper.*

**Suggested changes are implanted in the revision.**

3. *Line 177 – remove the word "months" after "July and August"*

**Suggested changes are implanted in the revision.**

4. *Line 187 – "CCN characteristics at the site are…"*

**Suggested changes are implanted in the revision.**

5. *Line 266 – "An enhancement in k-values is…"*

**Suggested changes are implanted in the revision.**

6. *Line 356 – should say "inactive"*

**Suggested changes are implanted in the revision.**

7. *Line 436 – "…are also similar…"*

**Suggested changes are implanted in the revision.**

8. *Line 565 – my name is misspelled. Should be Paramonov*
[Figure]

**Sorry for the mistake. Corrected**

**Suggested changes are implanted in the revision. In this regard we have modified Fig. 14 and more discussions as suggested is added in Line 585 as,**

**'The better association of CCN with absorption coefficient especially prior to the monsoon is interesting. The strong convective conditions existing during these conditions over the region can take the AP to high altitudes where it can absorb radiation and may lead to semi-direct effects. The association of BC with accumulation mode AP during continental conditions suggest that the carbonaceous AP existing in this size range can act as CCN. Hence, the role of carbonaceous AP in modulating both cloud microphysics and dynamics need to be investigated in detail. However, current investigation could not address these probable aspects.'**

**We didn't have enough evidence to show the role of dust aerosols influencing the CCN properties during the study period. As the reviewer mentioned in the 'minor comments', we have checked the possibility of dust coated with carbonaceous aerosols and found that possibility is absent. But we do agree that the point is valid over the region especially during the pre-monsoon to monsoon transition period and will be investigated using single particle and mixing state observations.**

*3. What hypothesis the authors put forward to explain the reduced activation ratio during monsoon/ marine airmass conditions, when normally the aerosols would be richer in hygroscopic species, that could be easily activated?*

**The reduced CCN activation ratio during the marine/wet conditions is due the removal of CCN active aerosol particles by wet scavenging and cloud activation. Similar observations of reduced CCN efficiency of AP due to the dominance of smaller AP are also observed over the Western Ghats during the monsoon (Jayachandran et al., 2018). This point is highlighted in the manuscript at Line 478 as,**

**'During the marine conditions, most of the bigger AP which are potential CCN are either washed out by wet scavenging or already activated as cloud droplets. Hence the measured AP are devoid of CCN active particles which are clearly seen from the aerosol NSD during the relevant periods (Fig. 11). Thus, the low CCN activation fraction during**

the marine conditions is due to the missing of those CCN active particles near the surface.'

*4. In CCN closure analysis, describe the methodology and assumptions used in estimating CCN. Why aerosol composition is assumed to be ammonium sulphate? In the analysis authors have explicitly tried to establish the influence of carbonaceous aerosols in acting as a CCN?*

**The details of CCN closure analysis are added in the Appendix.**

**Assumption of ammonium sulphate as the soluble fraction of the aerosol composition is the ideal case as sulfates are known to be the best CCN (Bigg, 1986; Covert et al., 1998). In the absence of concurrent aerosol composition measurements, we wanted to check the deviation of CCN activation from the ideal scenario. Hence ammonium sulfate was assumed as the aerosol composition following VanReken et al., (2003), Medina et al., (2007) etc. Though the study points to the pivotal role of carbonaceous AP toward CCN activation, we do not have quantitative information of aerosol composition in order to use in the CCN closure analysis.**

**Minor Comments**

*Line 32-40: Sentence is confusing and needs modification. It gives a feel that "Condensation nuclei" and "Cloud Condensation nuclei" are same. The sentence starting with "For a fixed liquid water content……." need to be revised.*

**Suggested changes are implanted in the revision. The Lines are modified as,**

**'Those AP or condensation nuclei (CN) which act as the Cloud Condensation Nuclei (CCN) at a specific supersaturation (SS) can indirectly affect the climate by altering the cloud micro-physical properties. In general, an increase in AP increases the cloud droplet concentration with smaller sizes (Twomey and Warner, 1967) for a fixed liquid water content, which in turn increases the cloud albedo (Twomey, 1977) and cloud lifetime (Albrecht. 1989).'**

*Line 40-45: "Characterization of CCN………………… the physical and chemical characteristics of AP". These two sentences can be reframed to one as it tries to convey the same information.*

**Suggested changes are implanted in the revision.  The lines are modified as,**

**'Characterization of the hygroscopic growth of AP, which is generally addressed by the Kohler theory (Kohler, 1936), is the most fundamental aspect in assessing the aerosol-cloud interactions (ACI) for reducing the uncertainties in indirect radiative forcing estimation.'**

*Line 52-55: "For a given particle...............the accuracy of climate models to address the ACI (Fountakis and Nenes,2005)". The relevance of this sentence in the paragraph is not understood. In the second paragraph authors try to portray the heterogeneities of aerosol particles and CCN in the global scenario as well as in Indian context. Authors can discuss more on the role of organics as CCN as they can reveal the first indirect effect (Nenes et al., 2002) and as well as studies over organics in Indian context.*

**Suggested changes are implanted in the revision. More discussions as suggested are added in the modified manuscript.**

*Line 73-77: "Various studies........ from the unique data obtained from the CAIPEEX". Rephrase the sentence.*

**Suggested changes are implanted in the revision. The lines are modified as,**

**'Various studies have addressed the spatio temporal distribution of AP (Padmakumari et al., 2013; Varghese et al., 2019), cloud microphysics (Prabha et al., 2011; 2012; Padmakumari et al., 2018), rainfall (Maheshkumar et al., 2014) properties, the relationship between cloud microphysics and thermodynamics (Bera et al., 2019), and ACI (Pandithurai et al., 2012; Prabha et al., 2012; Konwar et al., 2012; Gayatri et al., 2017; Patade et al., 2019) from the unique data obtained from the CAIPEEX.'**

*Line 80-90: Authors have mentioned that a few studies (Leena et al., 2016, Jayachandran et al., 2018) have already reported CCN characteristics over different locations of Western Ghats. If so, does this study address the same objectives with observations form a different site? Please bring more clarity to the objectives of this study.*

**The present study is the only study till date investigating the CCN properties at the leeward side of the Western Ghats, which is prone to drought conditions. Apart from the atmospheric dynamics, the role of aerosols in cloud nucleating over this region is still missing. Thus, the current study is the first attempt to understand the role of AP in cloud formation over the region. The main emphasis in the present study is the surface CCN observations during the Indian summer monsoon over the semi-arid rain shadow region, which has not been available from other observations**

*Line 110: Were the data corrected for the maximal activated fraction, which is of high importance, in particular for total CCN measurements (Paramonov et al., 2013; Rose et al., 2010)? Please give more information of reference data used in the köhler theory when performing the CCN calibration. This is very important because different parameterizations will retrieve different critical supersaturations (Rose et al.,2008; Wang et al., 2017). Also mention the uncertainty in measurements of different instruments.*

CCN counter was calibrated with ammonium sulfate aerosol following Rose et al. (2008). During the calibration experiments CCN efficiency spectra were recorded for different CCN column ΔT values. The activation diameter corresponding to 50% CCN efficiency for each spectrum was taken as the critical dry diameter for the CCN activation of ammonium sulfate particles. The corresponding critical supersaturation was calculated with the activity parameterization Köhler model (AP3) mentioned in Rose et al., (2008). The calculated critical supersaturation was taken as the effective supersaturation at the given ΔT value.

The CCN counter used in the study was factory calibrated at DMT Inc prior to the experiment and all the instruments are periodically calibrated during the experiment. The uncertainty associated with all the measurements are less than 10%.

These details are added in the manuscript.

*Line 162-165: Rewrite the captions specifically for Figure1. (a) & (b).*

Suggested changes are implanted in the revision.

*Line 186: Correct bullet numbering.*

Corrected.

*Line 190-220: There are several concerns in this analysis: (1) Does it mean that for all days in June and September, trajectories ending over study region were of continental origin? This is difficult to comprehend especially when authors have mentioned that monsoon onset over study region was on 08 June2018. (2) AOD retrieved from MODIS over land and especially during ISM is a matter of concern. (3) Authors also mentioned about MODIS retrieved fire count information, please do mention the confidence level used as well as its uncertainty.*

All the analysis in the current manuscript are based on the continental and marine airmass trajectories which coincides with the dry and wet conditions over the region, respectively, within the Indian Summer Monsoon period (June to September) of 2018. Though months were mentioned, it represents the corresponding days only which experienced the marine/continental air mass. However, to avoid the confusion the periods are uniformly referred as continental (1 and 2) and marine (1 and 2) conditions in the modified manuscript. Continental-1 and 2 comprise of 1-8 June and 15-30 September. While, the marine-1 and 2 comprises of 8-12, 15-31 of July and 1-26. 28-31 of August. These are mentioned in Table 1. However, considering the reviewer's concern, the details of the observation periods are added in the text as well. Single terminology (continental and marine) is implemented in the whole manuscript including the figures.

The fire spots are detected on daily basis from the MODIS sensors aboard the Terra and Aqua platforms at 1 X 1 km$^2$ spatial resolution globally. The detection is performed based

on the algorithm (Giglio et al., 2003; 2016) that uses the strong emission of mid-infrared radiation from fires. In the present study, we have considered only the data having confidence value higher than 30, which comes under the classification of 'nominal' and 'high'. This information is added to the manuscript.

*Line 225-270: From this study as well as those conducted over Mahabaleshwar and Amazon, an increase in ïA¸n´ values is reported during wet months. What is the scientific reason?*

**The increase in 'k' value during the wet months is associated with the enhancement in ultrafine particles which are CCN-inactive, in the total AP concentration. The increase of these small particles due to the wet scavenging necessitates higher supersaturation conditions for CCN activation. Hence, the CCN concentration increase drastically at high supersaturations resulting in a comparatively high 'k' value. The role of aerosol size in determining the k value is discussed in Nair et al., (2020).**

*Line 315-350: Justify your arguments on the formation of NPF over the study region during wet conditions? Why the CN-CCN relationship weakens during September? CN-CCN relationship seems to hold strong when CN concentration is _3.7\*10ˆ3 particles/cm-3. Is it due to instrumental artefact? Or do you propose any process?*

**The enhancement in nucleation mode particles (<30 nm) are seen mostly during the afternoon hours over the region. During the wet conditions, the AP concentration capable of providing deposition of pre-cursor gases are very less as seen in Figure 11, which can lead to the nucleation of AP. We have not investigated this aspect in detail in the current study. Hence new particle formation may be one of the reasons for the observed AP distribution having GMD less than 50 nm during the marine air mass.**

**There could be other reasons such as local emissions and turbulent mixing, which may be investigated later. Note that meteorology (PBL mixing) has a decisive role in the redistribution of particles of this size range.**

**From the closure studies, it can be seen that the accumulation mode particles (>100 nm) observed during continental-2 is less CCN active indicating the presence of hygophobic combination compared to continental-1. Also, from Fig. 11, it can be seen that the nucleation mode particles are more than twice during continental-2 than 1. Thus, both the presence of nucleation mode particles and lesser hygroscopic accumulation mode particles are responsible for the weak CN-CCN relationship during the September, compared to June.**

*Line 360: Figure 8 clearly shows that activation fraction (SS) is very low during wet months. What does it imply? Are similar observations are reported elsewhere? Discuss more on the implication and physical mechanisms?*

**Low CCN activation fraction during the wet periods are reported by Jayachandran et al., (2018) over the southern tip of peninsula. The concurrent AP number-size distribution (Fig. 11) points to the predominance of smaller particles due to wet scavenging during these conditions and are not getting replenished. The CCN-inactive particles are responsible for the low CCN activation fraction. This has a major impact on cloud microphysics which is a challenging task to quantify. The enhanced ultrafine AP concentration during the large supersaturation conditions can intensify the convective strength (Fan et al., 2018) and is an active area is further research.**

*Line 365-370: Why the diurnal variation in Twomey's empirical fit parameter –ïA̧n´ and activation fraction is not showing (revere) relationship that is obviously seen in other months. During September, diurnal variation in GMD showed morning high values, which is also reflected in AF but not in Twomey's empirical fit parameter –ïA̧n´. Similarly, in June, GMD showed low values during morning hours without any significant change in AF and Twomey's empirical fit parameter –ïA̧n´. Discuss the scientific implications of these*

**As the reviewer have rightly pointed out, the clear diurnal variations observed in GMD and activation fraction is not seen in the Twomey's empirical fit parameter-k, during September. The inverse relationship between CCN activation fraction and Twomey's empirical fit parameter- k fails during the presence of very small AP (Nair et al., 2020). There is a predominant nucleation mode AP during September (Fig. 11). The aerosol composition prior to and post the monsoon rainfall is a major missing factor in this analysis. These inferences point to the fact that the Twomey's empirical fit- k is not a perfect parameter to assess the CCN activation of AP. From Fig. 10, it can be seen that the diurnal variation in GMD during June is negligible (within the error bars) even though there is a dip during 10-16 hrs local time. Correspondingly there is no diurnal variation in CCN activation fraction and the 'k' value**.

*Line 395-400: Authors have tried to associate high Twomey's empirical fit parameter –ïA̧n´ values observed during morning hours of wet months to organic aerosol mostly produced by biomass burning. However, the MODIS fire count map during August shows very less fire count over the study region. Justify the statement?*

**As the reviewer rightly pointed out, the biomass burning indicated by the MODIS fire count is less over the study region during the wet period. Since the sampling site is on a rural setting, the use of solid fuels is dominant for domestic purposes which can contribute to the organic aerosols. Studies (Zhang et al., 2013) have shown that secondary organic carbons have high absorption Angstrom exponent.**

*Line 400-414: "Thus, the aerosol composition especially the organic aerosols: : :.." Please justify. In literature AAE greater than 2 is usually inferred as biomass source and AAE between 1 and 2 is usually*

*considered as mixture of BC and OC (Bergstorm et al., 2007). Authors should present independent observations or data to attribute this to organic aerosols.*

**Agree with the reviewer. AAE between 1 and 2 is due to the mixture of BC and OC and higher the AAE value point to the dominance of OC. The references cited in the corresponding discussion in the manuscript also indicate the same. Thus, in the present study, the sudden increase in 'k' values was associated with a high AAE, which indicate the predominance of OC. Due to the absence of aerosol composition measurements in the current study, it is difficult to attribute the exact type of organic aerosols causing this enhancement in both AAE and k values.**

*Line 515: Are the estimated 'a' and 'b' value are site and season specific. ?*

**The values are specific to the range of critical diameters which are season specific. However, the plot and the associated discussions are removed from the manuscript as per the other reviewer 1 suggestions.**

*Line 555: What about the role of dust aerosol (local/transported) acting as CCN? As can be seen in Figure, there are a few points where the AAE is less than 1. Some studies have attributed such points to dust coated with BC. Do back-trajectories in these case support dust transport?*

**We have looked the air mass trajectory for those cases. As it can be seen from the air mass back trajectories none of them are originating from the north-west or west part, from where dust transport is expected. To confirm the absence of dust over the region in this study, we have examined the aerosol NSD also for AAE<1 instance. Both the air mass back trajectories and NSD show negligible role of dust AP.**

*Line 590: Figure 13 and Figure 14 clearly indicates the role of carbonaceous aerosols acting as CCN over the study region during dry conditions (June and September). Is there any specific reason for the better correlation observed in June?*

**The co-occurrence of carbonaceous AP and CCN and associated relationship as seen in Fig. 13 and the weakening of the same during the September is noted. AP size plays a major role in this difference. A seen in Fig. 11, the NSD during the continental conditions in June has negligible nucleation mode particles (7%), compared to that of September (15%). This aspect is brought out more clearly by the modified Figure 14. The accumulation mode AP have a better correlation with the absorption coefficient during the dry condition prior to the monsoon. The atmospheric conditions will be favouring the aging of carbonaceous AP leading to the CCN activation. Detailed studies regarding the mixing state and size distribution of AP over the region is needed to confirm.**

*Straighten up the formatting errors in reference list. Check line 855 for example.*

**Suggested changes are implanted in the revision. All the references are now modified according to the journal norms**

[revised manuscript text omitted]